# Objective functions for information-theoretical monitoring network design: what is "optimal"?

Hossein Foroozand[1] and Steven V. Weijs[1]

[1]Department of Civil Engineering, University of British Columbia, Vancouver, British Columbia, Canada

**Correspondence:** Steven V. Weijs (steven.weijs@civil.ubc.ca)

**Abstract.** This paper concerns the problem of optimal monitoring network layout using information-theoretical methods. Numerous different objectives based on information measures have been proposed in recent literature, often focusing simultaneously on maximum information and minimum dependence between the chosen locations for data collection stations. We discuss these objective functions and conclude that a single objective optimization of joint entropy suffices to maximize the collection of information for a given number of stations. We argue that the widespread notion of minimizing redundancy, or dependence between monitored signals, as a secondary objective is not desirable and has no intrinsic justification. The negative effect of redundancy on total collected information is already accounted for in joint entropy, which measures total information net of any redundancies. In fact, for two networks of equal joint entropy, the one with a higher amount of redundant information should be preferred for reasons of robustness against failure. In attaining the maximum joint entropy objective, we investigate exhaustive optimization, a more computationally tractable greedy approach that adds one station at a time, and we introduce the "greedy drop" approach, where the full set of stations is reduced one at a time. We show that no greedy approach can exist that is guaranteed to reach the global optimum.

## 1 Introduction

Over the last decade, a large number of papers on information theory based design of monitoring networks have been published. These studies apply information-theoretical measures on multiple time series from a set of sensors, to identify optimal subsets. Jointly, these papers (Alfonso et al., 2010a, b; Li et al., 2012; Ridolfi et al., 2011; Samuel et al., 2013; Stosic et al., 2017; Keum and Coulibaly, 2017; Banik et al., 2017; Wang et al., 2018; Huang et al., 2020; Khorshidi et al., 2020) have proposed a wide variety of different optimization objectives. Some have suggested that either a multi-objective approach or a single objective derived from multiple objectives is necessary to find an optimal monitoring network. These methods were often compared to other existing methods in case studies used to demonstrate that one objective should be preferred over the other based on the resulting networks.

In this paper, we do not answer the question "what is optimal?" with an optimal network. Rather, we reflect on the question of how to define optimality in a way that is logically consistent and useful within the monitoring network optimization context, thereby questioning the widespread use of minimum dependence between stations as part of the objectives. In fact, we argue that minimizing redundancy is a redundant objective.

The objective of a hydrological monitoring network depends on its purpose, which can usually be framed as supporting decisions. The decisions can be relating to management of water systems, as for example considered by Alfonso et al. (2010a) or flood warning and evacuation decisions in uncontrolled systems. Also purely scientific research can be formulated as involving decisions to accept or reject certain hypotheses, focus research on certain aspects, or collect more data (Raso et al., 2018). In fact, choosing monitoring locations is also a decision, whose objective can be formulated as choosing monitoring locations to optimally support subsequent decisions.

The decision problem of choosing an optimal monitoring network layout needs an explicit objective function to be optimized. While this objective could be stated in terms of a utility function (Neumann and Morgenstern, 1953), this requires knowledge of the decision problem(s) at hand and the decision-makers preferences. These are often not explicitly available, for example, in the case of a government-operated monitoring network with a multitude of users. As a special case of utility, it is possible to state the objective of a monitoring network in terms of information (Bernardo, 1979). This can be done using the framework of information theory, originally outlined by Shannon (1948), who introduced information entropy H(X) as a measure of uncertainty or missing information in the probability distribution of random variable X, as well as many related measures.

Although ultimately the objective will be a more general utility, the focus of this paper is on information-theoretical methods for monitoring network design, which typically do not optimize for a specific decision problem supported by the network. Because information and utility (value of information) are linked through a complex relationship, this does not necessarily optimize decisions for all decision makers. Since we do not consider a specific decision problem, the focus in the present paper is on methods for maximization of information retrieved from a sensor network.

In this paper, the rationale behind posing various information-theoretical objectives is discussed in detail. While measures from information theory provide a strong foundation for mathematically and conceptually rigorous quantification of information content, it is important to pay attention to the exact meaning of the measures used. This paper is intended to shed some light on these meanings in the context of monitoring network optimization and provides new discussion motivated in part by recently published literature.

We present three main arguments in this paper. Firstly, we argue that objective functions for optimizing monitoring networks can, in principle, not be justified by analysing the resulting networks from application case studies. Evaluating performance of a chosen monitoring network would require a performance indicator which in itself is an objective function. Case studies could be helpful in assessing whether one objective function (the optimization objective) could be used as an approximation of another, underlying, objective function (the performance indicator). However, from results of case studies we should not attempt to draw any conclusions as to what objective function should be preferred. In other words: the objective function is intended to assess the quality of the monitoring network, as opposed to a practice where the resulting monitoring networks are used to assess the quality of the objective function. Secondly, we argue that, in purely information-based approaches, the joint entropy of all signals together is in principle sufficient to characterize information content and can therefore serve as single optimization objective. Notions of minimizing dependence between monitored signals through incorporation of other information measures in the objective function lack justification and are therefore not desirable.Thirdly, we could actually argue for maximizing redundancy as a secondary objective because of its associated benefits for creating a network robust against

individual sensors' failures. The reason is that the undesirable information inefficiencies associated with high dependency or redundancy are already accounted for in maximizing joint entropy. Minimization of redundancy would mean that each sensor becomes more essential, and therefore the network as a whole more vulnerable to failures in delivering information. Adding a trade-off with maximum redundancy is outside the scope of this paper, but serves to further illustrate the argument against use of minimum redundancy.

## 1.1 Choice of scope and role of the case study

The information-theoretical approach to monitoring network design is not the only option, and other objective functions have also been used for this problem. Examples are cost, geographical spread, and squared error based metrics. Some approaches use models describing spatial variability with certain assumptions, e.g. Kriging (Bayat et al., 2019). In the case of network expansion to new locations, models are always needed to describe what could be measured in those locations. These could vary from simple linear models to full hydrodynamic transport models, such as used in Aydin et al. (2019), who compared performance of the sensor placement in a polder network based on a simple low-order PCA (principal component analysis) model and a detailed hydrodynamic and salt transport model.

In this paper, our main focus is to discuss the formulation of information-theoretical objective functions and previous literature. Therefore, we restrict our scope to those information-theory based objective functions, based on observed data on one single variable in multiple locations. Keeping this limited scope allows us to discuss the interpretation of these objective functions for monitoring network design, which formalize what we actually want from a network. Furthermore, we investigate whether the desired optimum in the objective function can be found by greedy approaches, or whether exhaustive search is needed to prevent a loss of optimality.

Only after it is agreed on what is wanted from a network and this is captured as an optimization problem, other issues such as the solution or approximation of the solution to the problem become relevant. The numerical approach to this solution and calculation of information measures warrants independent discussion, which is outside our current scope and will be presented in a future paper. Our discussion is numerically illustrated by a case study using data from Brazos River in Texas, as presented in Li et al. (2012), to allow for comparisons. However, as we argue, the case study can only serve as illustration, and not for normative arguments for use of a particular objective function. Such an argument would be circular, as the performance metric will be one of the objective functions.

In this paper, since we are discussing the appropriate choice of objective function, there is no experimental setup that could be used to provide evidence for one objective version versus the other. Rather, we must make use of normative theoretical reasoning, and shining light on the interpretation of the objectives used and their possible justifications. The practical case studies in this paper therefore serve as illustration, but not as evidence for all the conclusions advocated in this paper, some of which are arrived at through interpretation and argumentation in the discussion section.

The manuscript is organized as follows. In the following methodology section, we introduce the methods used to investigate and illustrate the role of objective functions. In section 3, we discuss the case study on the streamflow monitoring network of Brazos River. Section 4 introduces the results for the various methods, and then discusses the need for multiple objectives, the

interpretation of trade-offs between redundancy and total information, and the feasibility of greedy algorithms reaching the optimum. The article concludes with summarizing the key messages and raising important questions about the calculation of the measures, to be addressed in future research.

## 2 Methodology

### 2.1 Information theory terms

Shannon (1948) developed information theory (IT) based on entropy, the concept that explains a system's uncertainty reduction as a function of added information. To understand how, consider a set of N events for which possible outcomes are categorized into m classes. Uncertainty is a measure of our knowledge about which outcome will occur. Once an event is observed, and it is identified which of the m classes it belongs to, our uncertainty about the outcome decreases to 0. Therefore, information can be characterized as the decrease of an observer's uncertainty about the outcome (Krstanovic and Singh, 1992; Mogheir et al., 2006; Foroozand and Weijs, 2017; Foroozand et al., 2018; Konapala et al., 2020). For monitoring networks, we are interested in the information content of the observations from all stations. The information content is equal to the uncertainty about the observations before measuring. The uncertainty is quantified through probability distributions that describe the possible observations, based on the data.

In monitoring network design, IT has been applied in the literature to evaluate data collection networks that serve a variety of purposes, including rainfall measurement, water quality monitoring, and streamflow monitoring. These evaluations are then used to optimize the placement of sensors. In the monitoring network optimization literature, three expressions from IT are often used in monitoring network design: (1) entropy (H), to estimate the expected information content of observations of random variables; (2) Mutual information, often called transinformation (T), to measure redundant information or dependency between two variables; (3) total correlation (C), a multivariate analogue to mutual information, to measure the total nonlinear dependency among multiple random variables.

The Shannon entropy H (X) is a nonparametric measure, directly on the discrete probabilities, with no prior assumptions on data distribution. It is also referred to as discrete marginal entropy, to distinguish it both from continuous entropy and from conditional entropy. Discrete marginal entropy (Eq.1), defined as the average information content of observations of a single discrete random variable X, is given by:

$$H(X) = -\sum_{x \in X} p(x) \log_2 p(x) \tag{1}$$

where $p(x)$ $(0 \leqslant p(x) \leqslant 1)$ is the probability of occurrence of outcome $x$ of random variable X. Equation 1 gives the entropy in the unit of "bits" since it uses a logarithm of base 2. The choice of the logarithm's base for entropy calculation is determined by the desired unit — other information units are "nats" and "Hartley" for the natural and base 10 logarithms, respectively. For monitoring network design, the logarithm of base 2 is common in the literature since it can be interpreted as the needed number of answers to a series of binary questions, and allows comparisons with file sizes in bits; see e.g. Weijs et al. (2013a). Joint

entropy (Eq.2), as an extension of entropy beyond a single variable, measures the number of questions needed to determine the outcome of a multivariate system.

$$H(X_1, X_2) = -\sum_{x_1 \in X_1} \sum_{x_2 \in X_2} p(x_1, x_2) \log_2 p(x_1, x_2) \tag{2}$$

where $p(x_1, x_2)$ is joint probability distribution of random variable $X_1$ and $X_2$. For a bivariate case, if two random variables are independent, then their joint entropy, $H(X_1, X_2)$, is equal to the sum of marginal entropies $H(X_1) + H(X_2)$. Conditional entropy (Eq.3), $H(X_1|X_2)$, explains the amount of information $X_1$ delivers that $X_2$ can not explain.

$$H(X_1|X_2) = -\sum_{x_1 \in X_1} \sum_{x_2 \in X_2} p(x_1, x_2) \log_2 \frac{p(x_1, x_2)}{p(x_2)} \tag{3}$$

$$0 \leqslant H(X_1|X_2) \leqslant H(X_1) \tag{4}$$

$H(X_1|X_2)$ can have a range (Eq.4) between zero when both variables are completely dependent and marginal entropy $H(X_1)$ when they are independent. Mutual information, in this field often referred to as transinformation (Eq.5), $T(X_1, X_2)$, explains the level of dependency and shared information between two variables by considering their joint distribution.

$$T(X_1; X_2) = -\sum_{x_1 \in X_1} \sum_{x_2 \in X_2} p(x_1, x_2) \log_2 \frac{p(x_1, x_2)}{p(x_1) * p(x_2)} \tag{5}$$

where $p(x_1)$ and $p(x_2)$ constitute the marginal probability distribution of random variable $X_1$ and $X_2$, respectively; and $p(x_1, x_2)$ form their joint probability distribution. The assessment of the dependencies beyond three variables can be estimated by the concept of Total Correlation (Eq.6) (proposed by McGill (1954) and named by Watanabe (1960)).

$$C(X_1, X_2, \ldots, X_n) = \left[ \sum_{i=1}^{n} H(X_i) \right] - H(X_1, X_2, \ldots, X_n) \tag{6}$$

Total Correlation (C) gives the amount of information shared between all variables by taking into account their nonlinear dependencies. C can only be non-negative since sum of all marginal entropies cannot be smaller than their multivariate joint entropy, though in the special case of independent variables, C would become zero.

## 2.2 Single-objective optimization

In this paper, we argue for the Maximum Joint Entropy (maxJE) objective for maximizing the total information collected by a monitoring network. This is equivalent to the GR3 objective proposed by Banik et al. (2017), as part of six other objectives (see appendix B for more detail) proposed in the same paper, which did not provide preference for its use. In the discussion, we argue that a single-objective optimization of the joint entropy of all selected sensors lead to a maximally informative sensor network, which minimizes total remaining uncertainty about the outcomes at all potential locations. Also, it should be

noted that the maxJE objective function already penalizes redundant information through its network selection process, which aims to find a new station that produces maximum joint entropy when it is combined with already selected stations in each iteration. When applied in a greedy search, adding one new station at a time, this approach ranks stations based on growing joint information as quickly as possible. This is mathematically equivalent to add to the selection, in each iteration, the new station $F_C$ that provides maximum conditional entropy $H(F_c|S)$ on top of an already selected set (S) of stations (see Figure 1 for visual illustration).

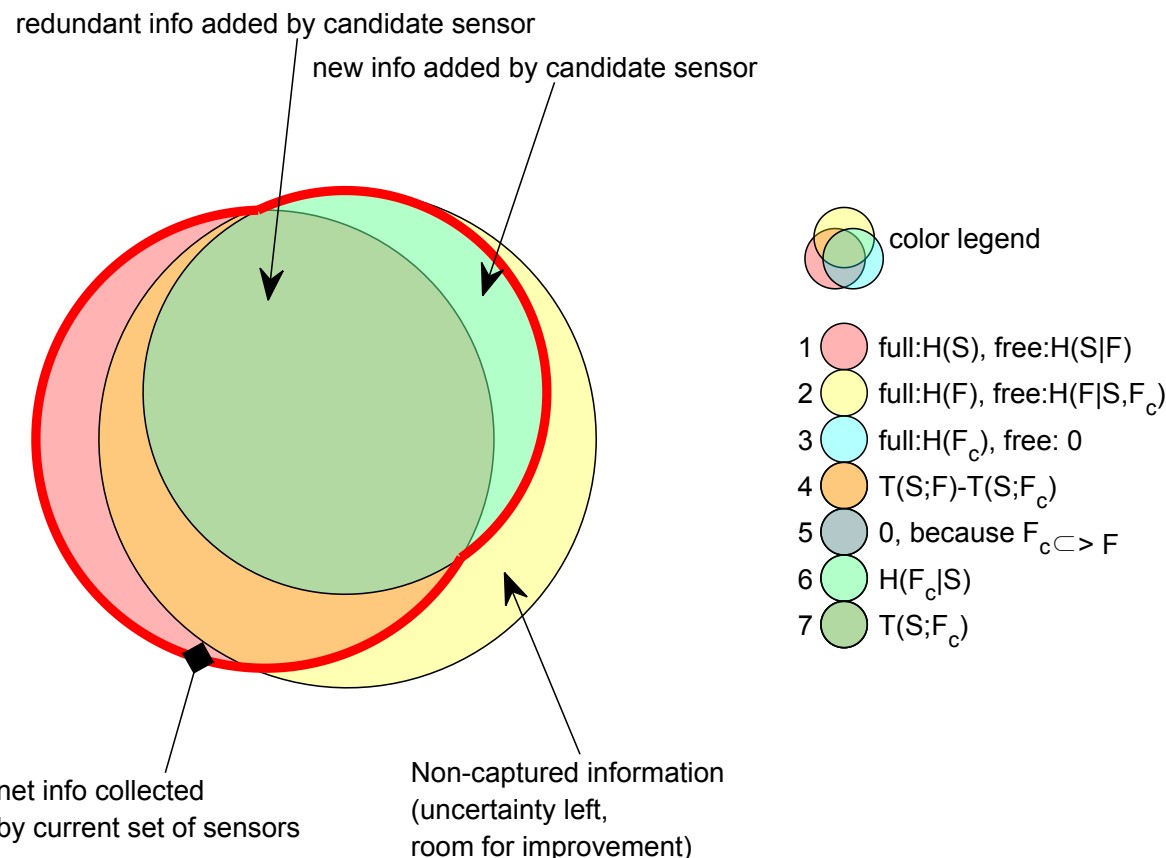

**Figure 1.** Venn diagram illustrating the relations between the relevant information measures. In the legend, the joint and marginal information-theoretical quantities (joint) entropy H(X), conditional entropy H(X|Y), and transinformation T (X;Y) for the variable from sets of already selected sensors S, not yet selected sensors F and the current candidate sensor $F_C$ are represented by the surfaces in the Venn diagram. For the 3 basic circle colors (first three circles in the legend), "free" gives the quantity represented by the non-covered part and "full" gives the quantity represented by the entire circle surface. The joint entropy that is proposed to be maximized in this paper is the area enclosed in the thick red line.

## 2.3 Multi-objective optimization

Our research compares and contrasts a variety of objective functions from literature. Information theory-based multi-objective optimization methods for monitoring networks have gained significant attention recently. Maximizing network information content, through either the sum of marginal entropy or joint entropy, is the common theme among existing methods (Alfonso et al., 2010b; Li et al., 2012; Samuel et al., 2013; Keum and Coulibaly, 2017; Wang et al., 2018; Huang et al., 2020). However, there is no consensus on whether to use total correlation or transinformation measures to minimize redundant information. Table 1 gives an overview of the large number of objectives and combinations of objectives used in the last decade. On the one hand, water monitoring in polders (WMP) method (Alfonso et al., 2010a) and joint permutation entropy (JPE) method (Stosic et al., 2017) used normalized transinformation to minimize redundant information. While, on the other hand, multi-objective optimization problem (MOOP) method (Alfonso et al., 2010b), Combined regionalization and dual entropy-multi-objective optimization (CRDEMO) method (Samuel et al., 2013), multivariable hydrometric networks (MHN) method (Keum and Coulibaly, 2017) and greedy rank based optimization (GR 5 and 6) method (Banik et al., 2017) adopted total correlation to achieve minimum redundancy. Interestingly, both C and T were used as competing objectives in maximum information minimum redundancy (MIMR) method proposed by Li et al. (2012). They argued that transinformation between selected stations in the optimal set and non-selected stations should be maximized to account for the information transfer ability of a network. Meanwhile, recently proposed methods in the literature attempted to improve monitoring network design by introducing yet other additional objectives (Huang et al., 2020; Wang et al., 2018; Banik et al., 2017; Keum and Coulibaly, 2017). These additional objective are further discussed in Appendix B.

## 2.4 Exhaustive search vs greedy add and drop

Apart from the objective function, the optimization of monitoring networks is also characterized by constraints. These constraints can either be implemented for numerical reasons or to reflect practical aspects of the real world problem. The majority of existing literature listed in Table 1 often implicitly imposed a constraint by treating stations' selection as greedy optimization. Greedy optimization adds one station to the selected stations each time, without reconsidering the set's already selected stations. A practical reason for this is numerical efficiency; an exhaustive search of all subsets of k stations out of n possible stations will need to consider a large number of combinations, since the search space grows exponentially with the size n of the full set of sensors ($2^n$ combinations of sensors need to be considered).

In this paper, for the maximization of joint entropy that we advocate, we consider and compare 3 cases for constraints with a large influence on computational cost, with the purpose of investigating whether these influence the results. We also interpret the constraints as reflections of placement strategies. Firstly, the "greedy add" strategy is the commonly applied constraint that each time the network expands, the most favorable additional station is chosen, while leaving the already chosen network intact. The optimal network for $k$ stations is found by expanding one station at a time. This approach can for example be useful in Alpine terrain, where relocating a sensor requires significant effort (Simoni et al., 2011). Secondly, "greedy drop" is the reverse strategy, not previously discussed in the literature, where the starting point is the full network with all $n$ stations. The

optimal network for $k$ stations is found by reducing the full network one step at a time, each step dropping the least informative station. Since all of the discussed monitoring design strategies use recorded data and hence discuss networks whose stations are already established, network reduction is perhaps the more realistic application scenario for information-based design methods. Thirdly, "exhaustive search" is the strategy where the optimal network of $k$ stations is found by considering all subsets of $k$ stations out of $n$. This unconstrained search is far more computationally expensive, and may not be feasible in larger networks for computational reasons. It can therefore be seen as a optimality benchmark. Because all options are considered, this is guaranteed to find the optimal combination of selected sensors for each network size, given the objective function.

In this comparison, we investigate whether the exhaustive optimization yields a series of networks where an increase in network size may also involve relocating stations. This may not always be practically feasible or desired in actual placement strategies, where networks are slowly expanded one station at a time. Occurrence of relocation in the sequence of growing subsets would also show that no greedy algorithm could exist that guarantees optimality.

## 2.5   Understanding and visualizing the measures of information

In this paper, we argue that, due to the additive relations of information measures (Eq.6), the proposed objectives functions in the literature are unnecessarily complicated, and a single-objective optimization of the joint entropy of all selected sensors will lead to a maximally informative sensor network. The additive relations between some of the information measures discussed in this paper are illustrated in Figure 1. In this figure and later is this paper, we use a shorthand notation: we use the sets of stations directly in the information measures, as a compact notation for the multivariate random variable measured by that set of stations. Various types of information interactions for three variables are conceptually understandable using Venn diagram (Figure 2.a). Although a Venn diagram can be used to illustrate information of more than three variables when they are grouped in three sets (Figure 1), it can't be used to illustrate pairwise information interactions beyond three variables. A chord diagram, on the other hand, can be useful to better understand pairwise information interaction beyond three variables. Figure 2 provides simple template to interpret and compare Venn and chord diagrams.

There are two important caveats with these visualizations. In the general Venn diagram of 3-variate interactions, the "interaction information", represented by the area where 3 circles overlap, can become negative. Hence, the Venn Diagram ceases to be an adequate visualization. For similar reasons, in the chord diagram, the sector size of outer arc lengths should not be interpreted as a total information transferred (Bennett et al., 2019). Information that can contribute to interactions is a combination of unique, redundant and synergistic components (Goodwell and Kumar, 2017; Weijs et al., 2018; Franzen et al., 2020). Their information entanglement is an active area of research in 3 or more dimensions. In this paper, the total size of the outer arc lengths is set to represent the sum of pairwise information interactions (used in Alfonso et al. (2010a)) and conditional entropy of each variable. This size may be larger than the total entropy of the variable and does not have any natural or fundamental interpretation.

In this paper, we use Venn diagrams to illustrate information relations between 3 groups of variables. Group one is the set of all sensors that are currently selected as being part of the monitoring network, which we denote as $S$. Group two is the set of all sensors that are currently not selected, denoted as $F$, and group 3 is the single candidate sensor that is currently considered

for addition to the network, $F_c$; see appendix A for an overview of notation. Since group 3 is a subset of group 2, one Venn circle is contained in the other, and there are only 5 distinct areas vs 7 in a general 3-set Venn diagram. In this particular case, there is no issue arising from negative interaction information.

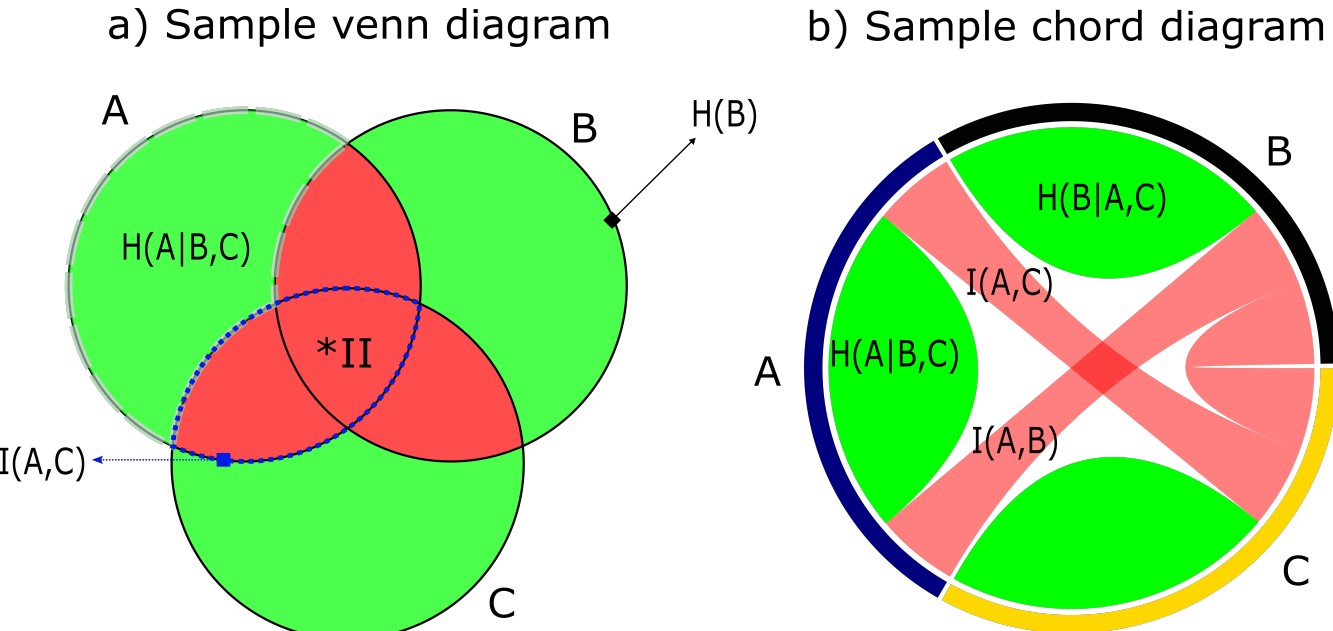

**Figure 2.** Template illustrations of information interactions with (a) Venn diagram, (b) chord diagram. The green and red areas in both diagrams show a graphical representation of conditional entropy and mutual information respectively. The solid line circles in Venn diagram depict single-variable entropy. *II is information interaction between three variables. The sector size in the outer circle in chord diagram is composed of arcs whose relative lengths correspond to the sum of pairwise information interactions and conditional entropy of each variable, and are best not interpreted.

### 2.6  Objective functions used in comparison for this study

For the purpose of illustrating the main arguments of this study, we compare maxJE objective function (Eq.7) with three other

(sets of) objective functions from previously proposed methods: MIMR (Eq.8), WMP (Eq.9) and minT (Eq.10). These methods were chosen since they are highly cited methods in this field, and more importantly, recent new approaches in the literature have mostly been built on one of these methods with additional objectives Alfonso et al. (2010a, b); Ridolfi et al. (2011); Li et al. (2012); Samuel et al. (2013); Stosic et al. (2017); Keum and Coulibaly (2017); Banik et al. (2017); Wang et al. (2018); Huang et al. (2020).

Objective function (maxJE): $= maximize\ \mathrm{H}\left(\langle X_{S_1}, X_{S_2}, \ldots, X_{S_k}\rangle, X_{F_C}\right)$         (7)

$$
\text{Objective function (MIMR):} = \begin{cases}
maximize\ \mathrm{H}\left(\langle X_{S_1}, X_{S_2}, \ldots, X_{S_k}\rangle, X_{F_C}\right) \\
maximize\ \sum_{i=1}^{m} \mathrm{T}\left(\langle X_{S_1}, X_{S_2}, \ldots, X_{S_k}\rangle, X_{F_i}\right) \\
minimize\ \mathrm{C}\left(\langle X_{S_1}, X_{S_2}, \ldots, X_{S_k}\rangle, X_{F_C}\right) \\
\therefore \blacktriangledown integrated\, format \blacktriangledown \\
maximize\ \lambda_1(\mathrm{H}\left(\langle X_{S_1}, X_{S_2}, \ldots, X_{S_k}\rangle, X_{F_C}\right) + \sum_{i=1}^{m} \\
\mathrm{T}\left(\langle X_{S_1}, X_{S_2}, \ldots, X_{S_k}\rangle, X_{F_i}\right)) - (1 - \lambda_1)\mathrm{C}\left(\langle X_{S_1}, X_{S_2}, \ldots, X_{S_k}\rangle, X_{F_C}\right)
\end{cases}
\tag{8}
$$

$$
\text{Objective function (WMP):} = \begin{cases}
maximize\ \mathrm{H}(F_C) \\
subject\ to\ \sum_{i \in S} \frac{\mathrm{T}(S_i; F_C)}{\mathrm{H}(S_i)} < SBM
\end{cases}
\tag{9}
$$

$$
\text{Objective function (minT):} = \begin{cases}
maximize\ first\ \mathrm{H}(F_C) \\
minimize\ \mathrm{T}\left(\langle X_{S_1}, X_{S_2}, \ldots, X_{S_k}\rangle, X_{F_C}\right)
\end{cases}
\tag{10}
$$

Where $\langle X_{S_1}, X_{S_2}, \ldots, X_{S_k}\rangle$ refers to selected stations in the previous iterations. $\mathrm{X}_{F_C}$ and $\mathrm{H}(F_C)$ denote the variable at the current candidate station and its marginal entropy, respectively. $F_C$ is the station considered for addition to the current set in a greedy-add approach. This formulation was chosen to allow a uniform presentation between methods. The objectives

for methods maxJE and MIMR can easily be modified to consider $F_C$ as part of $S$, so that the objective function evaluates the entire network rather than one candidate station for addition. This allows greedy-add, greedy-drop and exhaustive search methods. $\lambda_1$ is information-redundancy trade-off weight (Li et al., 2012). $SBM$ (select below median) stands for constraint where only stations are considered that are below the median score of all potential stations on that objective. $m$ is equal to the number of non-selected station in each iteration ($m + k = n$ total number of stations). For the multi-objective approaches

used in the case study, we used the same weights as the original authors to identify a single solution. It can be seen that a large number of different combinations of information-theoretical measures are used as objectives.

## 3 Study area and data description

In previous studies, the focus of the research has been on finding an optimal network for the subject case study with only little discussion on the theoretical justification of applying a new methodology. For this reason, the primary goal of this paper is

255 critically discussing the rationale for use of several objective functions in monitoring network design. To illustrate differences between the methods, we decided to apply our methodology to the Brazos River streamflow network (Figure 3) since this network was subject of study for the MIMR method. This network is under-gauged, according to the World Meteorological Organization density requirement. However, using the exact same case study eliminates the effect of other factors besides the objective function on the comparison. Such factors could be initial network density, temporal, and spatial variability. To isolate

our comparison from those effects, as well as from methodological choice such as resolution, time period considered, and

quantization method, we used the same data period and floor function quantization (Eq.11) proposed by Li et al. (2012).

$$x_q = a \left\lfloor \frac{2x + a}{2a} \right\rfloor \tag{11}$$

Here, $a$ is the histogram bin-width for all intervals except the first one, for which the bin-width is equal to $\frac{a}{2}$. $x$ is station's streamflow value, and $x_q$ is its corresponding quantized value; and $\lfloor \ \rfloor$ is the conventional mathematical floor function.

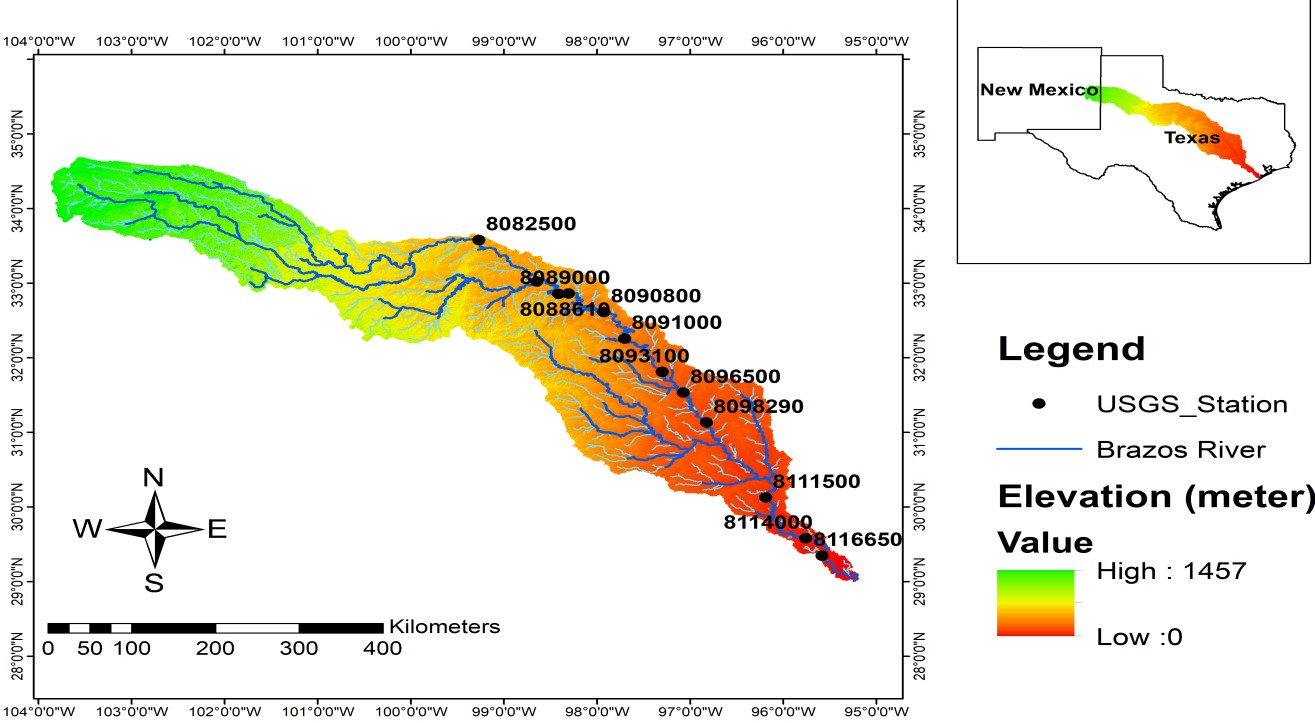

**Figure 3.** Brazos streamflow network and USGS stream gauges locations.

In Li et al. (2012), 12 USGS stream gauges on the Brazos River were selected for the period of 1990-2009 with monthly temporal resolution; some statistics of the data are presented in Figure 4. For the discretization of the time series, they used a binning approach where they empirically optimized parameter $a$ to satisfy three goals: (1) to guarantee all 12 stations have distinguishable marginal entropy, (2) to keep spatial and temporal variability of stations' time series, bin-width should be fine enough to capture the distribution of the values in the time series while being coarse enough so that enough data points are
available per bin to have a representative histogram, and (3) to prevent rank fluctuation to due to the bin-width assumption, sensitivity analysis must be conducted. They carried out the sensitivity analysis and proposed $a = 150 \ m^3/s$ for this case study, the resulting marginal entropy for each station is illustrated in Figure 4.

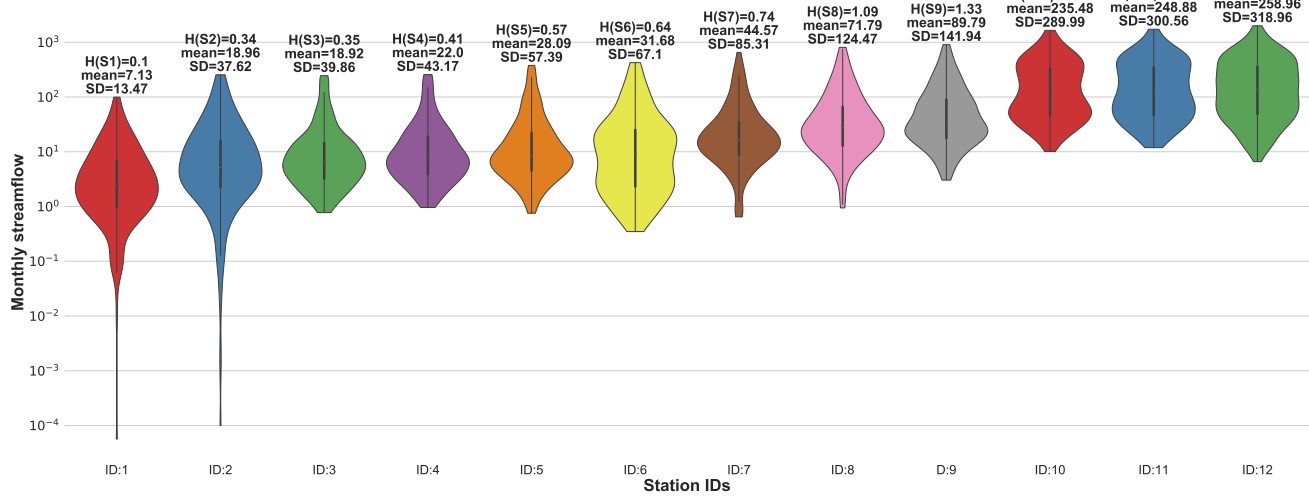

**Figure 4.** Brazos River streamflow $(m^3/s)$ statistics and resulting entropy values (bits). The stations' IDs are organized from upstream to downstream gauges in the watershed. Entropy values are calculated by floor function and parameter $a = 150\ m^3/s$.

## 4 Results and Discussion

### 4.1 Comparison of the objectives for Brazos River case

As indicated in the introduction, we should not attempt to gauge the merits of the objective functions by the intuitive optimality of the resulting network. Rather, the merits of the networks should be gauged by the objective functions. Still, the case study can provide insight in some behaviours resulting from the objective functions.

To assess and illustrate the workings of the different objectives in retrieving information from the water system, we compared three existing methods with a direct maximization of the joint entropy of selected sensors, $\mathrm{H}(S, \mathrm{F}_c)$, indicated with maxJE in

the results, such as Tables 2 and 3. The joint entropy results in Table 2 indicate that maxJE is able to find a combination of 8 stations that contains joint information of all 12 stations ranked by other existing methods. Figure 5 displays spatial distribution of the top 8 stations chosen by different methods. Before any interpretation of the placement, we must note that the choices made in quantization and the availability of data play an important role in the optimal networks identified. Whether the saturation that occurs with 8 stations has meaning for the real world case study depends on whether the joint probability

distribution can be reliably estimated. This is highly debatable and merits a separate detailed discussion which is out of the scope of this paper. We present this case study solely to illustrate behaviour of the various objectives.

**Table 2.** Resulting maximum joint entropy (bits) for different number of gauges found with different methods for Brazos River case study (JE used exhaustive optimization)

| Method | Multivariate dimensions | | | | | | | | | | | |
|---|---|---|---|---|---|---|---|---|---|---|---|---|
| | 1 | 2 | 3 | 4 | 5 | 6 | 7 | 8 | 9 | 10 | 11 | 12 |
| MIMR | 2.47 | 2.84 | 2.87 | 3.21 | 3.23 | 3.23 | 3.23 | 3.32 | 3.33 | 3.52 | 3.93 | 4.1 |
| WMP1/2 | 2.47 | 3.07 | 3.21 | 3.36 | 3.38 | 3.38 | 3.38 | 3.38 | 3.38 | 3.52 | 3.82 | 4.1 |
| minT | 2.47 | 2.53 | 2.69 | 2.72 | 2.76 | 2.89 | 3.06 | 3.08 | 3.33 | 3.52 | 3.93 | 4.1 |
| maxJE | 2.47 | 3.07 | 3.5 | 3.7 | 3.88 | 4.02 | 4.09 | 4.1 | 4.1 | 4.1 | 4.1 | 4.1 |

Note that MIMR's trade-off weight ($\lambda_1 = 0.8$) is based on the recommendation of Li et al. (2012) for this dataset.

**Table 3.** Optimal gauge orders found with different methods for Brazos River case study.

| Method | Station ranking in multivariate dimensions | | | | | | | | | | | |
|---|---|---|---|---|---|---|---|---|---|---|---|---|
| | 1 | 2 | 3 | 4 | 5 | 6 | 7 | 8 | 9 | 10 | 11 | 12 |
| MIMR | 12 | 6 | 1 | 8 | 2 | 3 | 4 | 7 | 5 | 9 | 10 | 11 |
| WMP1/2 | 12 | 9 | 7 | 6 | 5 | 4 | 3 | 2 | 1 | 8 | 11 | 10 |
| minT | 12 | 1 | 2 | 3 | 4 | 5 | 7 | 6 | 8 | 9 | 10 | 11 |
| maxJE | 12 | 9 | 10 | 11 | 8 | 5 | 7 | 2* | 1* | 3* | 4* | 6* |

Note that for the last 5 stations, indicated with *, multiple optimal orders are possible.

The most notable difference between MaxJE and the other methods is the selection of all 3 of the stations located most downstream. While other methods would not select these together due to high redundancy between them, maxJE still selects all stations, because despite the redundancy, there is still found to be enough new information in the second-most and third-most downstream station. This can be in part attributed to the quantization choice of equally sized bins throughout the network, leading to higher information contents downstream. While this quantization choice is debatable, it is important, in our opinion, to not compensate artifacts from quantization by modifying the objective function, even if the resulting network may seem more reasonable, but rather to address those artifacts in the quantization choices themselves. To repeat the key point: An objective function should not be chosen based on whether it yields a "reasonable network" but rather based on whether the principles that define it are reasonable.

Though already necessarily true from the formulation of the objective functions, we use the case study to illustrate how other methods with a separate minimum redundancy objective lead to the selection of stations with lower new information content (green area in Figure 6). Reduction of the yellow area in each iteration (i.e. the information loss compared to the full

network) in Figure 6 corresponds to the growth of joint entropy values in Table 2 for each method. maxJE (by definition) has the fastest, and minT the slowest rate of reduction of information loss. Methods' preference for reaching minimum redundancy or growing joint information (red area in Figure 6) governs the reduction rate of information loss. Also, Figure 7 provides auxiliary information about the evolution of pairwise information interaction between already selected stations $X_1, X_2, \ldots, X_{i-1}$ in the previous iterations and new proposed station $X_i$. Figure 7 illustrates the contrast between the choice of the proposed stations in the first six iterations by different methods. For instance, minT method aims to find a station that has minimum mutual information (red links in Figure 7) with already selected stations. In contrast, the maxJE method tries to grow joint entropy, which translates to finding a station that has maximum conditional entropy (green segments in Figure 7). Other methods opt to combine two approaches by either imposing a constraint (WMP) or having a trade-off between them (MIMR).

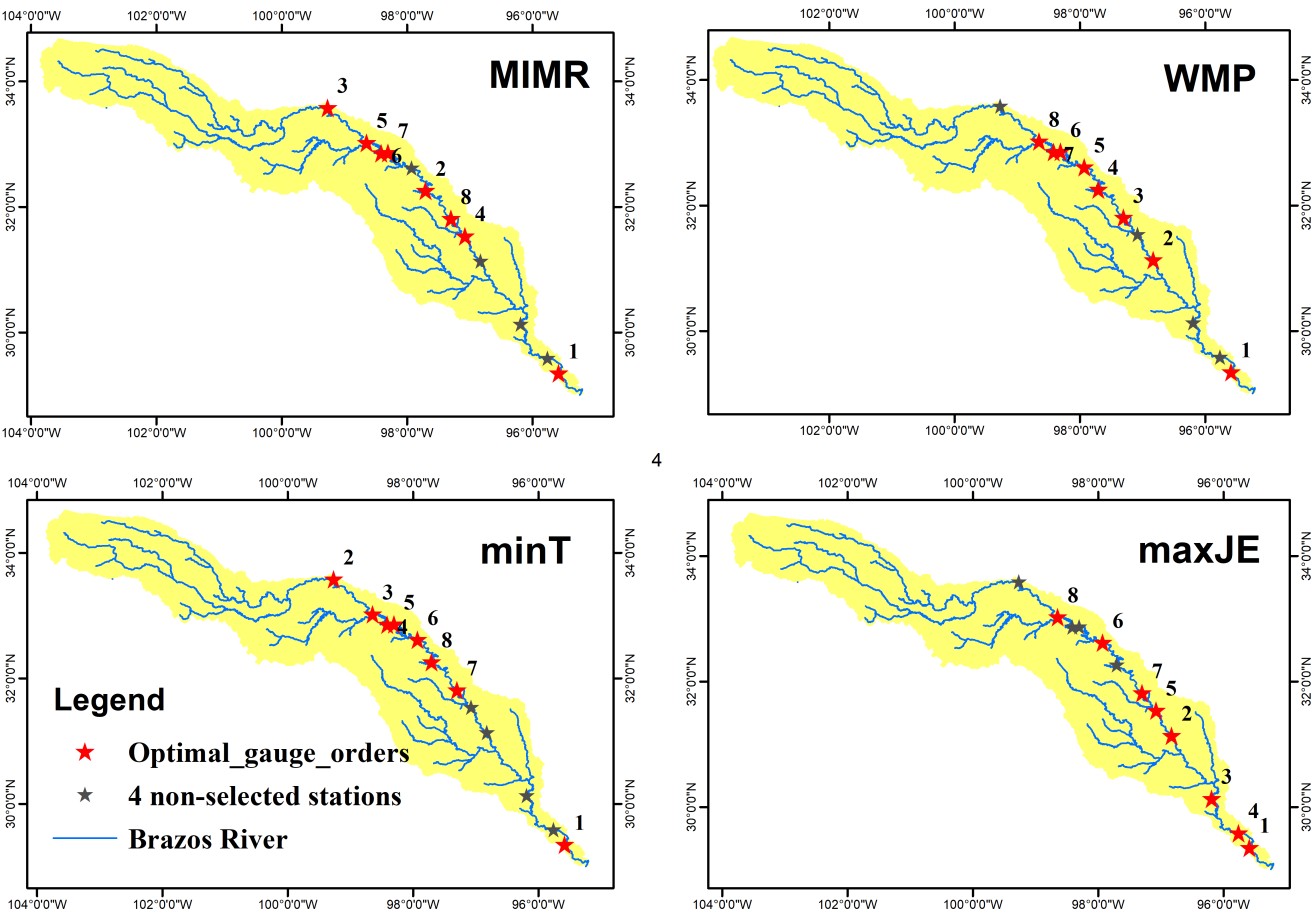

**Figure 5.** Spatial distribution of the top 8 streamflow gauges ranked different objectives.

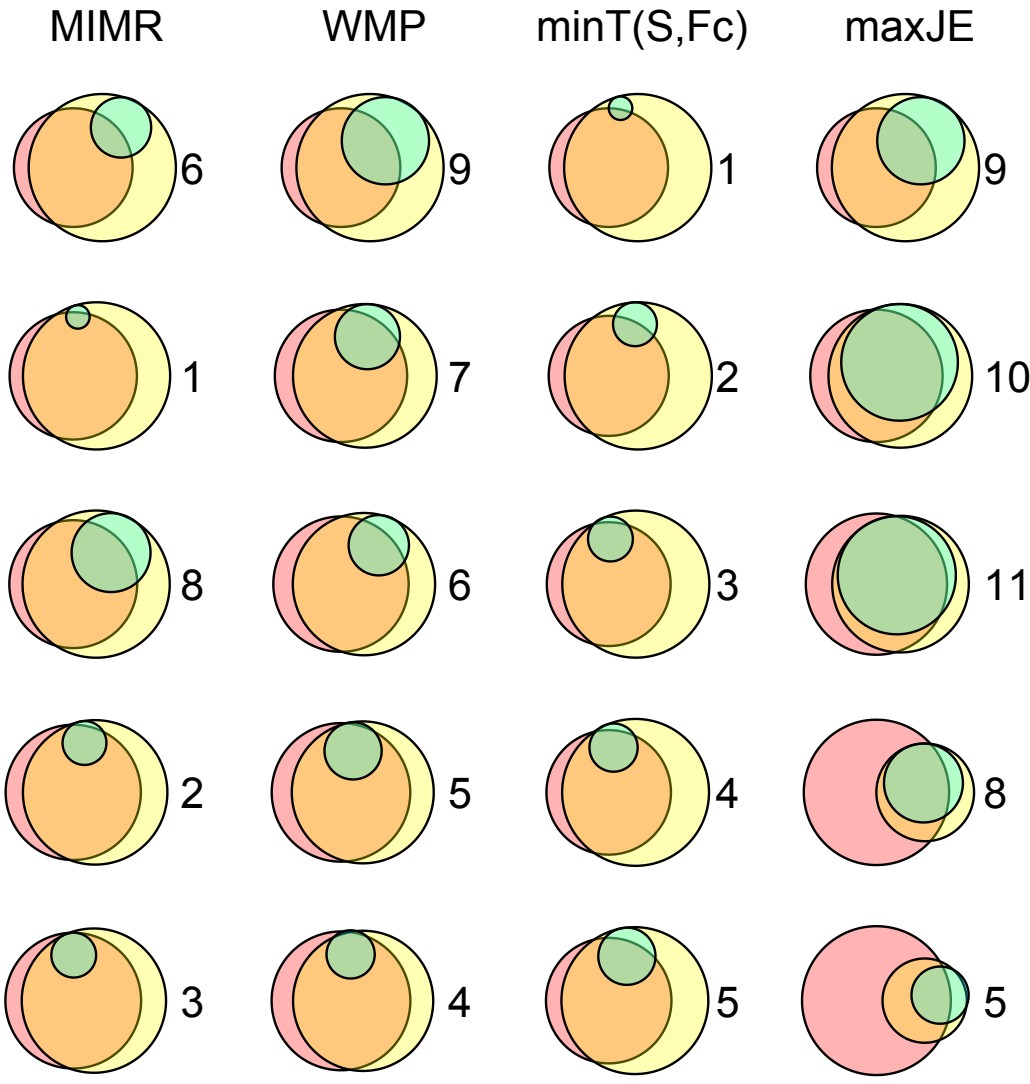

**Figure 6.** Approximately proportional Venn diagrams showing the evolution of information measures when progressively (going downwards on the rows) selecting stations (selected station for each step indicated by the numbers) using four different methods (in the different columns). The interpretation of the color-coded areas representing the information measures is the same as in figure 1. All methods select station 12 as the initial station (entropy given by pink circle on row 1). As can be seen from the diagram on the bottom right, the method maximizing joint entropy leaves almost no information unmeasured (yellow part) with just 6 stations, while the other methods still miss capturing this information. Exact numbers behind the Venn diagram can be found with the code available with this paper.

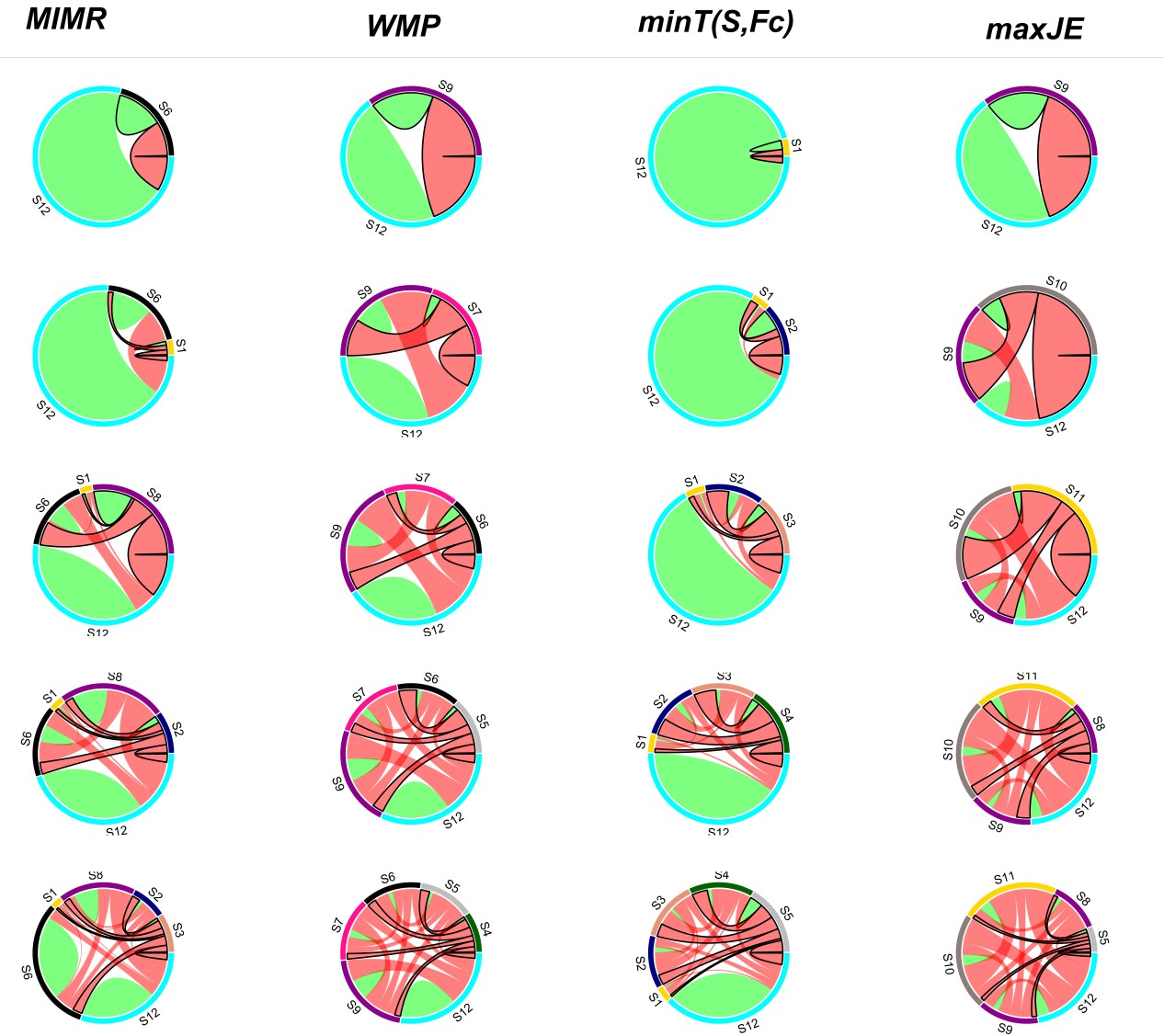

**Figure 7.** Evolution of pairwise information interaction between already selected stations in the previous iterations and new proposed station. Green and red links represent proportional conditional entropy and mutual information,respectively. Links with black border emphasizes on the information interaction of new proposed station in each iteration.

## 4.2 Is minimization of dependence needed?

The existing approaches considered above have in common that they all involve some form of dependence criterion to be minimized. For example, the total correlation gives a measure of total redundant information within the selected set. This is information that is duplicated and therefore does not contribute to the total information content of the sensors, which is given by the joint entropy. Focusing fully on minimizing dependence, such as done in the minT objective optimization, makes the

optimization insensitive to the amount of non-duplicated information added. This results in many low entropy sensors being selected. It is important to note that the joint entropy already accounts for duplicated information and only quantifies the non-redundant information. This is exactly the reason why it is smaller then the sum of individual entropies. In terms of joint entropy, two completely dependent stations are considered to be exactly as informative as one of them. This means that the negative effect that dependency has on total information content is already accounted for by maximizing joint entropy only.

Mishra and Coulibaly (2009) stated that "The fundamental basis in designing monitoring networks based the entropy approach is that, the stations should have as little transinformation as possible, meaning that the stations must be independent of each other". However, no underlying argument for this fundamental basis is given in the paper. The question is then whether there is another reason, apart from information maximization, why the total correlation should be minimized. In three of the early papers (Alfonso et al., 2010a, b; Li et al., 2012) introducing the approaches that employed or evaluated total correlation, no such reason was given other than the one by Mishra and Coulibaly (2009). Also in later citing research (Huang et al., 2020; Wang et al., 2018; Keum and Coulibaly, 2017; Stosic et al., 2017; Fahle et al., 2015), no such arguments have been found except for effectiveness, which we argue is covered by looking at the total non-redundant information the network delivers. Traditional reasons for minimizing redundancy are reducing the burden of data storage and transmission, but these are not very relevant in monitoring network design, since those costs are often negligible compared to the costs of the sensor installation and maintenance (see (Barrenetxea et al., 2008; Nadeau et al., 2009; Simoni et al., 2011)). Moreover, information theory tells us that, if needed, redundant information can be removed before transmission and storage by employing data compression (Weijs et al., 2013b, a). The counter-side of minimal redundancy is less reliability, a far more relevant criterion for monitoring network design. Given that sensors often fail or give erroneous values, one could argue that redundancy (total correlation) should actually be maximized, given a maximum value of joint entropy. We might even want to gain more robustness at the cost of losing some information. One could for example imagine placing a new sensor directly next to another to gain confidence in the values and increase reliability, instead of using it to collect more informative data in other locations.

The Pareto front that would be interesting to explore in this context is the trade-off between *maximum* total correlation (robustness) vs. joint entropy (expected information gained from the network), indicated by the red line in Figure 8. Different points on this Pareto front reflect different levels of trust in the sensors' reliability. Less trust requires more robustness and leads to a network design yielding more redundant information. Previous approaches, such as the MOOP approach proposed by Alfonso et al. (2010b), explore the Pareto front given by the black dashed line, where *minimum* total correlation is conflicting with maximizing joint entropy. As argued in this section, this trade-off is not fundamental in information-theoretical terms. Still, it results from the fact that there is usually redundant information as a by-product of new information, so highly informative stations also carry more redundant information. This redundant information does not reduce the utility of the new information, so does not need to be included as a minimization objective in the optimization.

Summarizing, the maximization of joint entropy while minimizing redundancy is akin to maximizing effectiveness while maximizing a form of efficiency (i.e. bits of unique information / bits collected). However, bits do not have any significant associated cost. If installing and maintaining a monitoring location has a fixed cost, then efficiency should be expressed as

unique information gathered per sensor installed, which could be found by maximizing joint entropy (effectiveness) for a given number of stations, as we suggest in this paper.

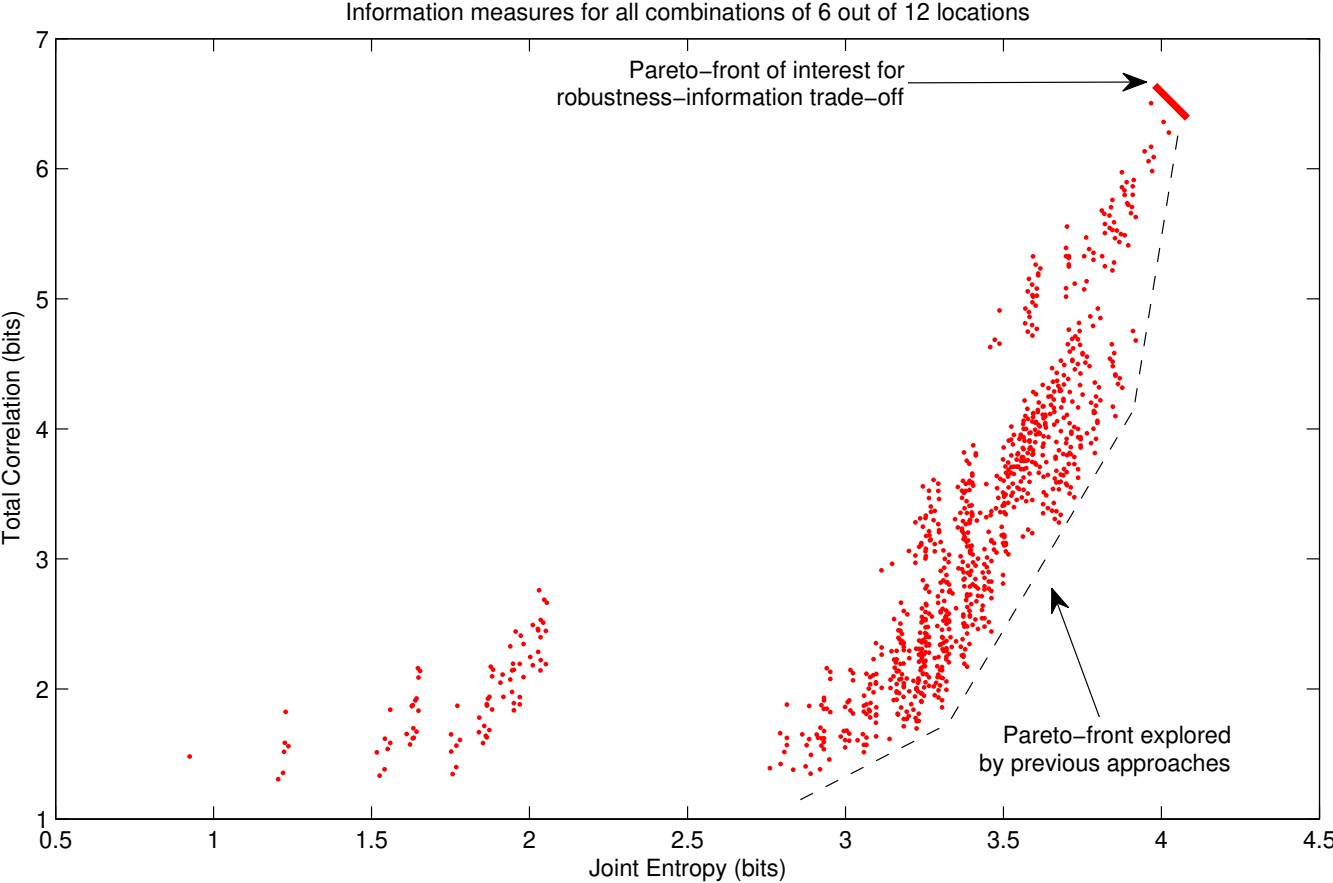

**Figure 8.** The resulting total correlation and joint entropy for all 924 possible combinations of 6 out of 12 sensor locations. In some past approaches, a Pareto front in the lower right corner is given importance. In this paper, we argue that this trade off is irrelevant, and information can be maximized with the horizontal direction only. If a trade-off with reliability needs to be considered, the Pareto front of interest is in the top-right corner instead of the lower right corner that is previously recommended in the literature.

### 4.3    Greedy algorithms vs. exhaustive optimization of maximum joint entropy

Different search strategies have been adopted in the literature for monitoring network design. The most commonly used greedy algorithms impose a constraint on exhaustive search space to reduce computational effort. We investigated three different search strategies to obtain the optimal network in the context of using maxJE as an objective function. We discuss the advantages and limitations of each search strategy in terms of optimality of the solution and computational effort.

The exhaustive optimization tests all possible new combinations, not restricted to those combinations containing the already
selected set in a smaller network. Since the joint entropy of a set of locations does not depend on the order in which they are

added, the number of possible combinations is $\binom{n}{k}$ (i.e. $n$ choose $k$), where $n$ is the number of potential stations in the pool and $k$ is the number of selected stations. The computational burden is therefore greatest when about half of the stations are selected. For a number of potential sensors under 20, this is still quite tractable (4 minutes on normal PC, implemented -by a hydrologist- in MATLAB), but for larger numbers, the computation time increases very rapidly. When considering all sub-network sizes, the number of combinations to consider is $2^n$, so an exponential growth. We could make an optimistic estimate, only considering the scaling from station combinations to evaluate, but not considering the dimensionality of the information measures. For 40 stations, this estimate would yield a calculation time of more than 5 years, unless a more efficient algorithm can be found. Regardless of potential improvements in implementation, the exponential scaling will cause problems for larger systems.

Greedy approaches might be candidates for efficient algorithms. For the proposed joint entropy objective, we tested the optimality of greedy approaches against the benchmark of exhaustive optimization of all possible station combinations. For the Brazos River case study, both the "add" and "drop" greedy selection strategies resulted in the global optimum sets, i.e. the same gauge order and resulting joint entropy as was found by the exhaustive optimization. These results can be read from last row of Tables 2 and 3. Therefore, for this case, the greedy approaches did not result in any loss of optimality. For the last few sensors, multiple different optimal sets could be identified, which are detailed in Table 4. Results in Table 4 show multiple network layouts with equal network size and joint information exist. For this case, network robustness could be an argument to prefer the network with maximum redundancy.

In a further test, using artificially generated data, we experimentally falsified the hypothesis that greedy approaches can guarantee optimality. For this test, we generated a correlated random Gaussian dataset for 12 stations, based on the covariance matrix of the data from the case study. We increased the number of generated observations to 860 time steps, to get a more reliable multidimensional probability distribution. Table 5 shows the resulting orders for twelve stations for the three different approaches. Note how for the exhaustive optimization in this example, in some instances, one or two previously selected gauges are dropped in favor of selecting new stations. The resulting joint entropies for the selected sets are shown in Table 6. This means no greedy approach can exist that finds results equivalent to the exhaustive approach.

**Table 4.** All optimal combinations of sensors for the joint entropy objective. For number of sensors above 7, multiple optimal combinations can be found due to saturation of joint entropy. Black squares are selected sensors.

| Number of selected stations with multiple optimal combinations | Station ID | | | | | | | | | | | |
|---|---|---|---|---|---|---|---|---|---|---|---|---|
| | 1 | 2 | 3 | 4 | 5 | 6 | 7 | 8 | 9 | 10 | 11 | 12 |
| 1 | □ | □ | □ | □ | □ | □ | □ | □ | □ | □ | □ | ■ |
| 2 | □ | □ | □ | □ | □ | □ | □ | □ | ■ | □ | □ | ■ |
| 3 | □ | □ | □ | □ | □ | □ | □ | □ | ■ | ■ | □ | ■ |
| 4 | □ | □ | □ | □ | □ | □ | □ | □ | ■ | ■ | ■ | ■ |
| 5 | □ | □ | □ | □ | □ | □ | □ | ■ | ■ | ■ | ■ | ■ |
| 6 | □ | □ | □ | □ | ■ | □ | □ | ■ | ■ | ■ | ■ | ■ |
| 7 | □ | □ | □ | □ | ■ | □ | ■ | ■ | ■ | ■ | ■ | ■ |
| 8 | □ | ■ | □ | □ | ■ | □ | ■ | ■ | ■ | ■ | ■ | ■ |
| 8 | □ | □ | ■ | □ | ■ | □ | ■ | ■ | ■ | ■ | ■ | ■ |
| 9 | ■ | ■ | □ | □ | ■ | □ | ■ | ■ | ■ | ■ | ■ | ■ |
| 9 | ■ | □ | ■ | □ | ■ | □ | ■ | ■ | ■ | ■ | ■ | ■ |
| 9 | □ | ■ | ■ | □ | ■ | □ | ■ | ■ | ■ | ■ | ■ | ■ |
| 9 | □ | ■ | □ | ■ | ■ | □ | ■ | ■ | ■ | ■ | ■ | ■ |
| 9 | □ | ■ | □ | □ | ■ | ■ | ■ | ■ | ■ | ■ | ■ | ■ |
| 9 | □ | □ | ■ | ■ | ■ | □ | ■ | ■ | ■ | ■ | ■ | ■ |
| 9 | □ | □ | ■ | □ | ■ | ■ | ■ | ■ | ■ | ■ | ■ | ■ |
| 10 | ■ | ■ | ■ | □ | ■ | □ | ■ | ■ | ■ | ■ | ■ | ■ |
| 10 | ■ | ■ | □ | ■ | ■ | □ | ■ | ■ | ■ | ■ | ■ | ■ |
| 10 | ■ | ■ | □ | □ | ■ | ■ | ■ | ■ | ■ | ■ | ■ | ■ |
| 10 | ■ | □ | ■ | ■ | ■ | □ | ■ | ■ | ■ | ■ | ■ | ■ |
| 10 | ■ | □ | ■ | □ | ■ | ■ | ■ | ■ | ■ | ■ | ■ | ■ |
| 10 | □ | ■ | ■ | ■ | ■ | □ | ■ | ■ | ■ | ■ | ■ | ■ |
| 10 | □ | ■ | ■ | □ | ■ | ■ | ■ | ■ | ■ | ■ | ■ | ■ |
| 10 | □ | ■ | □ | ■ | ■ | ■ | ■ | ■ | ■ | ■ | ■ | ■ |
| 10 | □ | □ | ■ | ■ | ■ | ■ | ■ | ■ | ■ | ■ | ■ | ■ |
| 11 | ■ | ■ | ■ | ■ | ■ | □ | ■ | ■ | ■ | ■ | ■ | ■ |
| 11 | ■ | ■ | ■ | □ | ■ | ■ | ■ | ■ | ■ | ■ | ■ | ■ |
| 11 | ■ | ■ | □ | ■ | ■ | ■ | ■ | ■ | ■ | ■ | ■ | ■ |
| 11 | ■ | □ | ■ | ■ | ■ | ■ | ■ | ■ | ■ | ■ | ■ | ■ |
| 11 | □ | ■ | ■ | ■ | ■ | ■ | ■ | ■ | ■ | ■ | ■ | ■ |
| 12 | ■ | ■ | ■ | ■ | ■ | ■ | ■ | ■ | ■ | ■ | ■ | ■ |

**Table 5.** Resulting station selections for the artificially permuted dataset with 860 data points

| Method | | Station selection for various network sizes | | | | | | | | | | | |
|---|---|---|---|---|---|---|---|---|---|---|---|---|---|
| | | 1 | 2 | 3 | 4 | 5 | 6 | 7 | 8 | 9 | 10 | 11 | 12 |
| Exhaustive | Added | 3 | 1;12 | 6 | 5;7;11 | 2;6 | 9;12 | 3 | 10 | 5 | 8 | 4 | 11 |
| | Removed* | | 3 | | 6; 12 | 5 | 11 | | | | | | |
| Greedy Add | | 3 | 11 | 1 | 7 | 6 | 9 | 2 | 10 | 5 | 8 | 12 | 4 |
| Greedy Drop | | 1 | 12 | 6 | 7 | 2 | 9 | 3 | 10 | 5 | 8 | 4 | 11 |

∗ means a previously selected station is removed from optimal set of selected stations.

**Table 6.** Resulting joint entropy for the artificially permuted dataset with 860 data points

| Method | Number of stations | | | | | | | | | | | |
|---|---|---|---|---|---|---|---|---|---|---|---|---|
| | 1 | 2 | 3 | 4 | 5 | 6 | 7 | 8 | 9 | 10 | 11 | 12 |
| Exhaustive | 1.538 | 3.003 | 4.225 | 5.058 | 5.732 | 6.172 | 6.515 | 6.695 | 6.832 | 6.933 | 7.024 | 7.083 |
| Greedy add | 1.538 | 2.898 | 4.164 | 5.043 | 5.681 | 6.111 | 6.486 | 6.646 | 6.789 | 6.899 | 6.996 | 7.083 |
| Greedy drop | 1.530 | 3.003 | 4.225 | 5.041 | 5.724 | 6.172 | 6.515 | 6.695 | 6.832 | 6.933 | 7.024 | 7.083 |

Based on our limited case study, the questions remain open: 1) whether faster algorithms can be formulated that yield guaranteed optimal solutions, and 2) in which cases the greedy algorithm provides a close approximation. It is also possible to formulate modified greedy methods with the ability of replacing a limited number of stations instead of just adding stations. This leads to a significantly reduced computational burden, while reaching the optimum more often than when adding stations one at a time. In Table 5, it can be seen that allowing a maximum number of two relocated stations would already reach the

optimal configurations for this specific case. Another limitation of this comparison is that we did not consider metaheuristic search approaches (Deb et al., 2002; Kollat et al., 2008), which fall in between greedy and exhaustive approaches in terms of computational complexity, could serve to further explore the optimality versus computational complexity trade-off. It would be interesting to further investigate what properties in the data drive the sub-optimality of greedy algorithms. Synergistic interactions (Goodwell and Kumar, 2017) are a possible explanation, although our generated data example shows that even

when moving from 1 to 2 selected stations, a replacement occurs. Since there are only pairs of variables involved, synergy is not needed in the explanation of this behaviour. Rather, the pair with maximum joint entropy does not always include the station with maximum entropy, which could perhaps be too highly correlated with other high entropy variables.

## 5 Conclusions

The aim of this paper was to contribute to better understanding the problem of optimal monitoring network layout using information-theoretical methods. Since using resulting networks and performance metrics from case studies to demonstrate that one objective should be preferred over the other would be circular, the results from our case study served as an illustration of the effects, but not as arguments supporting the conclusions we draw about objective functions. We investigated the rationale for using various multiple-objective and single-objective approaches, and discussed the advantages and limitations of using exhaustive vs. greedy search. The main conclusions for the study can be summarized as follows:

- The purpose of the monitoring network governs which objective functions should be considered. When no explicit information about users and their decision problems can be identified, maximizing the total information collected by the network becomes a reasonable objective. Joint entropy is the only objective needed to maximize retrieved information, assuming that this joint entropy can be properly quantified.

- We argued that the widespread notion of minimizing redundancy, or dependence between monitored signals, as a secondary objective is not desirable and has no intrinsic justification. The negative effect of redundancy on total collected information is already accounted for in joint entropy, which measures total information net of any redundancies.

- When the negative effect on total information is already accounted for, redundant information is arguably beneficial, as it increases robustness of the network information delivery when individual sensors may fail. Maximizing redundancy as an objective secondary to maximizing joint entropy could therefore be argued for, and trade-off between these objectives could be explored depending on the specific case.

- The comparison of exhaustive and greedy search approaches shows that no greedy approach can exist that is guaranteed to give the true optimum subset of sensors for each network size. However, the exponential computational complexity, which doubles the number of sensor combinations to evaluate with every sensor added, makes exhaustive search prohibitive for large networks, illustrated by the following. During the COVID-19 response in March 2020, Folding@home, currently the world's largest distributed computing project, broke the exaFLOP barrier ($10^{18}$ floating point operations per second). Even with that computational power, it would take more than 10 years to evaluate a network of 90 potential stations, under the impossibly optimistic assumption that evaluating one network were possible in one FLOP. The complexity of the greedy approach is quadratic in the number of locations, and therefore feasible for large search spaces.

- The constraints to the search space imposed by the greedy approach could also be interpreted as a logistical constraint. In a network expansion scenario, it disallows replacement of stations already selected in the previous iteration.

- We introduced the "greedy drop" approach that starts from the full set and deselects stations one by one. We have demonstrated that the two types of greedy approaches do not always lead to the same result, and neither approach guarantees the unconstrained true optimal solution. Synergistic interactions between variables may play a role, although this is not the only possible explanation. In our case study, the suboptimality of greedy algorithms was not visible in

original data, but we demonstrated its existence with artificially generated data. In our specific case studies, differences between exhaustive and greedy approaches were small; especially when using a combination of the greedy add and greedy drop strategy. It remains to be demonstrated in further research how serious this loss of optimality is in a range of practical situations, and how results compare to intermediate computational complexity approaches such as metaheuristic algorithms.

## 5.1 Further work

In this paper, we focused on the theoretical arguments for justifying the use of various objective functions, and compared a maximization of joint entropy to other methods, while using the same data set and quantization scheme. Since the majority of previous research used greedy search tools to find optimal network configurations, we compared greedy and exhaustive search approaches to raise awareness in the scientific community that greedy optimization might fall into local optimum, though its application can be justified considering computation cost of exhaustive approach. Banik et al. (2017) compared computation cost for greedy and metaheuristic optimization (Non-dominated Sorting Genetic Algorithm II). They reported that the greedy approach resulted in drastic reduction of the computational time for the same set of objective functions (metaheuristic computation cost was higher 58 times in one trial and 476 times in another). We recommend further investigation of these three search tools in terms of both optimality (for the maxJE objective) and computation cost.

Another important question that needs to be addressed in future research is to investigate how the choices and assumptions made (i.e., data quantization which influences probability distribution) in the numerical calculation of objective functions would affect network ranking. What many of these objective functions have in common, is that they rely on multivariate probability distributions. For example, in our case study, the joint entropy is calculated from a 12-dimensional probability distribution. These probability distributions are hard to reliably estimate from limited data, especially in higher dimensions, since data requirements grow exponentially. Also, these probability distributions and the resulting information measures are influenced by multiple factors, including choices about the data's temporal scale and quantization. To have an unbiased comparison framework of objective functions, we kept data and quantization choices from a case study previously described in the literature. It is worth acknowledging that these assumptions, as well as data availability, can greatly influence station selection, and require more attention in future research.

Numerically, the limited data size in the case study presents a problem for the calculation of multivariate information measures. Estimating multivariate discrete joint distributions exclusively from data requires quantities of data that exponentially grow with the number of variables, i.e. potential locations. When these data-requirements are not met and joint distributions are still estimated directly based on frequencies, independent data will be falsely qualified as dependent and joint information content severely underestimated. This can also lead to apparent earlier saturation of joint entropy, at a relatively low number of stations. For the case study presented here, we do not recommend interpreting this saturation as reaching the number of needed stations, since it could be a numerical artifact. This problem applies to all methods discussed in this paper. Before numerics can be discussed, clarity is needed on the interpretation and choice of the objective function. In other words, before thinking about how to optimize, we should be clear on what to optimize. We hope that this paper helped illuminate this.

## Appendix A: Notation and definitions

| | |
|---|---|
| $S$ | Set of indices of selected stations locations |
| $F$ | Set of indices of potential monitoring locations not yet selected |
| $F_C$ | The index of the monitoring station currently under consideration for addition |
| $X_S, X_F, X_{F_C}$ | The (sets of) time series (variables) measured at the station(s) in the respective sets |
| $p(x_1)$ | The marginal probability distribution of random variable $X_1$ |
| $p(x_1, x_2)$ | The joint probability distribution of variable $X_1$ and $X_2$ |
| $H(F_C)$ | A shorthand for $H(X_{F_C})$. In information measures, the set is used as shorthand for variables in that set |
| $H(X_{F_C})$ | The entropy of the marginal distribution of time series in $F_C$ |
| $H(X_F|X_S)$ | The conditional joint entropy of variables in $F$, given knowledge of variables in $S$ |
| $T(X_F; X_S)$ | Mutual information or transinformation between set of variables in $F$ and set of variables in $S$ |
| $C(X_1, X_2, \ldots, X_n)$ | Total Correlation, the amount of information shared between all variables |
| $\lambda_1$ | Information-redundancy trade-off weight |
| $SBM$ | "select below median", the constraint where only stations are considered that are below the median score of all po |
| $a$ | Histogram bin-width |
| $x$ | Station's streamflow value |
| $x_q$ | Quantized value after discretization |
| AE | Apportionment entropy |
| RDI | Ranking disorder index |
| $SI_z(X)$ | Local spatiotemporal information of the grid $X$ in local window $z$ in the time series |
| $p(\sigma_z)$ | Probability distribution of the standard deviation $\sigma_z$ in time series |
| $A_{network}$ | Network accuracy |
| $Var$ | Kriging variance |
| $D$ | Detection time |
| $D_{sp}(\gamma)$ | The average of the shortest time among the detection times for monitoring station |
| $R$ | Reliability |
| $\delta_s$ | Binary choice of 1 or 0 for whether the contamination is detected or not |

## Appendix B: Additional objectives used in recent literature

Recent literature has expanded the information-theoretical objectives with additional objectives. For instance: (1) Wang et al. (2018) proposed dynamic network evaluation framework (DNEF) method that follows MIMR method for network configuration in different time windows and optimal network ranking is determined by maximum Ranking disorder index (RDI) (Eq.B2), which is normalized version of apportionment entropy (AE). RDI was proposed by Fahle et al. (2015) and named by Wang

et al. (2018) to analyze the uncertainty of the rank assigned to a monitoring station under different time windows; (2) Huang et al. (2020) proposed information content, spatiotemporality, and accuracy (ISA) method, which extends MIMR method by adding two objectives: maximizing spatiotemporality information (SI), and maximizing accuracy (A). The SI (Eq.B4) objective is introduced to incorporate spatiotemporality of satellite data into network design, and A (Eq.B5) objective is proposed to Maximize the interpolation accuracy of the network by minimizing the regional Kriging variance; (3) Banik et al. (2017) proposed six combinations (GR 1-6) of four objectives: detection time (D) (Eq.B6), reliability (R) (Eq.B7), H (Eq.2) and C (Eq.6) for locating sensors in sewer systems; and (4) Keum and Coulibaly (2017) proposed to maximize conditional entropy as a third objective in dual entropy-multi-objective optimization to integrate multiple networks (in their case: raingauge and streamflow networks). Although maximizing conditional entropy can indirectly be achieved in other used-objective (joint entropy), this new objective gives more preference to maximizing unique information that one network can provide when another network can't deliver. These multi-objective optimization problems are solved by either finding an optimal solution in a Pareto front (Alfonso et al., 2010b; Samuel et al., 2013; Keum and Coulibaly, 2017) or by merging multiple objectives with weight factors into a single objective function (Li et al., 2012; Banik et al., 2017; Stosic et al., 2017).

$$\text{AE} = -\sum_{i=1}^{n} \frac{r_i}{M} \log_2 \frac{r_i}{M} \tag{B1}$$

$$\text{RDI} = \text{nAE} = \frac{\text{AE}}{\log_2 n} \tag{B2}$$

Where $n$ is the number of possible ranks that a station can have (i.e., $n$ is equal to the total number of stations). $\frac{r_i}{M}$ ratio is an occurrence probability of the outcome, where $M$ is the number of ranks under different time windows, and $r_i$ is the number of a certain $i^{\text{th}}$ rank. Therefore, AE takes on its maximum value when the ranking probability of a station has equally probable outcome while minimum AE happens when the station's rank is constant. RDI ranges from 0 to 1, and higher RDI values indicate ranking sensitivity of a station to temporal variability of the data.

$$\text{SI}_z(X) = -\sum_{i=1}^{l} p(\sigma_z) \log_2 p(\sigma_z) \tag{B3}$$

$$\text{SI}_{network}(X, \gamma_{F_i}) = \frac{1}{n+1} \left[ \sum_{j=1}^{n} SI_z(X_{S_j}) + SI_z(\gamma_{F_i}) \right] \tag{B4}$$

$$\text{A}_{network}(X, \gamma_{F_i}) = -\frac{1}{l} \sum_{i=1}^{l} \sum_{j=1}^{k} Var_{ij} \tag{B5}$$

Where $\text{SI}_z(X)$ is the local spatiotemporal information of the grid $X$ in local window $z$ in the time series; and $p(\sigma_z)$ is probability distribution of the standard deviation $\sigma_z$ in time series $l$. $\text{SI}_{network}(X, \gamma_{F_i})$ is spatiotemporal of the network, which

is calculated by the average of spatiotemporal information of already selected sites $SI_z(X_{S_j})$ and a potential site $SI_z(\gamma_{F_i})$. $A_{network}(X, \gamma_{F_i})$ is network accuracy, and $Var$ is Kriging variance over time series $l$ and number of grids $k$ in the study area.

$$495 \quad D(\gamma) = \frac{1}{S} \sum_{s=1}^{S} D_{sp}(\gamma) \tag{B6}$$

$$R(\gamma) = \frac{1}{S} \sum_{s=1}^{S} \delta_s \tag{B7}$$

Where $S$ is the total number of scenarios considered, and $D_{sp}(\gamma)$ is the average of the shortest time among the detection times for monitoring stations, and $\delta_s$ is binary choice of 1 or 0 for whether the contamination is detected or not.

*Code and data availability.* Data and code availability

The code and data that were used to generate the results in this manuscript are available from https://github.com/hydroinfotheory and the USGS https://waterdata.usgs.gov/nwis.

*Author contributions.* SW conceptualized study, HF and SW jointly performed analysis and wrote manuscript. SW supervised HF

*Competing interests.* The authors declare no competing interests

*Acknowledgements.* This research was supported by funding from Hossein Foroozand's NSERC CGS - Doctoral award and Steven V. Weijs's NSERC discovery grant.

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

**Table 1.** Various information-theoretical objectives used by methods proposed in recent literature.

| Method | Reference | Objective Function | | | | | | | | |
|---|---|---|---|---|---|---|---|---|---|---|
| | | H($F_C$) | SBM | H($S,F_C$) | C($S$) | T($F_C;S$) | C($F_C;S$) | PE($F_C;S$) | H($Q|P$) | Others |
| **WMP1** | Alfonso et al. (2010a) | max | $W_1$ | | | | | | | |
| **WMP2** | Alfonso et al. (2010a) | max | $W_2$ | | | | | | | |
| **WMP3** | Alfonso et al. (2010a) | max | $W_3$ | | | | | | | |
| **MOOP** | Alfonso et al. (2010b) | | | max | | | min | | | |
| **minT** | Ridolfi et al. (2011) | max $1^{st}$ | | | | min | | | | |
| **MIMR** | Li et al. (2012) | | | $\lambda_1$ | $-(1-\lambda_1)$ | $\lambda_1$ | | | | |
| **CRDEMO** | Samuel et al. (2013) | | | max | | | min | | | |
| **JPE** | Stosic et al. (2017) | | | | | min | | max | | |
| **MHN** | Keum and Coulibaly (2017) | | | max | | | min | | max | |
| **GR1** | Banik et al. (2017) | | | | | | | | | min D |
| **GR2** | Banik et al. (2017) | | | | | | | | | max R |
| **GR3** | Banik et al. (2017) | | | max | | | | | | |
| **GR4** | Banik et al. (2017) | | | | | | | | | min D & max R |
| **GR5** | Banik et al. (2017) | | | max | | | min | | | |
| **GR6** | Banik et al. (2017) | | | max | | | min | | | min D & max R |
| **DNEF** | Wang et al. (2018) | | | $\lambda_1$ | $-(1-\lambda_1)$ | $\lambda_1$ | | | | max RDI |
| **ISA** | Huang et al. (2020) | | | $\lambda_1$ | $-(1-\lambda_1)$ | $\lambda_1$ | | | | max SI & max A |
| **maxJE** | this paper | | | max | | | | | | |

WMP objectives: $W_1 = \sum_{i \in S} \mathrm{T}(S_i; F_C)$, $W_2 = \sum_{i \in S} \frac{\mathrm{T}(S_i; F_C)}{\mathrm{H}(S_i)}$, $W_3 = \sum_{i \in S} \frac{\mathrm{T}(S_i; F_C)}{\mathrm{H}(F_C)}$

The table shows whether an objective is maximized (max) or minimized (min) or forms part of a weighted objective function that is maximized with weights $\lambda_1$. SBM stands for constraint where only stations are considered that are below the median score of all potential stations on that objective. D is detection time, and R is reliability. RDI stands for ranking disorder index. SI is spatiotemporality information, and A is accuracy. WMP=Water Monitoring in Polders; MOOP = Multi Objective Optimization Problem; minT = Minimum Transinformation; MIMR = Maximum Information Minimum Redundancy; CRDEMO = Combined Regionalization and Dual Entropy-Multi-objective Optimization; JPE = Joint Permutation Entropy; MHN = Multivariable Hydrometric Networks; GR = Greedy Rank; DNEF = Dynamic Network Evaluation Framework; ISA = information content, spatiotemporality and accuracy ; and maxJE = Maximum Joint Entropy.