# Peer review of "Objective functions for information-theoretical monitoring network design: what is optimal?"

_Hydrology and Earth System Sciences, 2020_

## Referee Comment (RC1) · Heye Bogena (Referee) · 29 Apr 2020

This paper deals with the optimisation of monitoring networks using information-theoretical methods with a focus on the analysis of useful objective functions. The authors argue that a single-objective optimization of the joint entropy of all selected sensors will lead to a maximally informative sensor network. They compare exhaustive optimization, a greedy approach and a new "greedy drop" approach using available monthly runoff data.

I enjoyed reading the manuscript, especially the introduction to the information theory terms. This study adds some interesting new views on the optimisation of monitoring
networks and fits well to the scope of HESS. However, although the paper is mostly well written, it is sometimes poorly structured. In addition, I have some general and specific remarks that needs to be considered before publication. I therefore recommend a minor revision of the paper.

General comments:

For this study, monthly runoff data were used from gauging stations with very different catchment areas. It can be assumed that the gauging stations with smaller catchment areas will show a higher temporal variability of discharge. Therefore, their sub-monthly data should have a higher information value than the data of the stations with larger catchments. Thus, the authors should also discuss the dependence of their results on the temporal scale of the discharge data used.

The order of presentation of the tables and figures is sometimes confused.

Specific comments:

L1: "layout"

L11-83: Subtitles within the introductory chapter are rather uncommon.

L123-145: Please always give the equations directly after their first citation.

L166: You are first referring to Appendix B instead of A. Should be reversed.

L184: Which part of the Appendix?

L243: Eq. 12 and the corresponding variable description should directly follow this sentence.

L259: "Tables"

L259-260: Please indicate the location of the eight stations containing the joint information of all 12 stations in Figure 3 and discuss whether the result is meaningful, e.g. in terms of the distance between stations, their respective catchment areas, etc..

L260-265: Results shown in Figs. 5 and 6 need to be explained in more detail.

L305-309: Repetition

L315-319: This estimate is of limited value because it depends largely on the programming code (i.e. Matlab is very slow compared to e.g. FORTRAN).

L323: "Tables"

L325: Table 6 deserves more explanation. The numbering of the tables is confusing. This should be rather Table 4.

L328-329: If find Table 4 difficult to understand. In addition, the captions of Table 4 and 5 indicate 240 data points, which should be rather 860*12 data points, if I understand correctly. I suggest combining Table 4 and 5.

L339: "generated"

L344-356: This section is more like introduction and discussion and thus not appropriate for a concluding chapter.

L357-363: You must present the most important results of your study more clearly, e.g. by using also bullet points.

L365-379: After removing any redundancies this section should be moved to the discussion section.

---

## Referee Comment (RC2) · Anonymous Referee #2 · 17 Jun 2020

This paper discussed the objective functions in information theory-based hydrometric network design problems and suggested a backward greedy approach instead of other optimization methods. While I was interested in reading this paper because taking more correct, reasonable, and meaningful objective functions and proper optimization techniques is very important in the network design using information theory, unfortunately, I couldn't get the answer of the question in the title, "what is optimal?", throughout this paper.

1. The authors argue that a higher amount of redundant information should be preferred because it reinforces the robustness of the network. However, this is not a
general statement but can be applied only for a specific condition. World Meteorological Organization has recommended the minimum network densities, which do not represent the optimal number of stations, rather they are suggested to avoid any serious deficiency in water resources management. That is, just meeting the guideline couldn't be sufficient while there is no clear definition of sufficient density. However, many regions in the world, such as developing countries, do not meet even the WMO guideline. Even in the study area in this paper, Brazos River Watershed, the drainage area is about 116,000 sq.km. and there are 12 USGS stations considered as the existing stations. The current network density becomes 9,667 sq.km per station while the minimum network density of the WMO guideline for the Interior Plains is 1,875 sq.km., which is more than five times sparser network. Considering the WMO's minimum network is the baseline not to lose critical information, it should be noted that the network is seriously under-gauged, such that we need to consider network expansion and network efficiency rather than network robustness. In this case, minimizing total correlation is more meaningful to optimize network efficiency. Besides, installing and maintaining monitoring stations often highly depend on financial budgets which cannot be satisfactory in practice. On the other hand, if the network density largely exceeds the minimum network density, and the water resources managers consider shrinking network by closing stations, the authors' argument may become agreeable.

2. The authors also argue that maximizing joint entropy already connotes minimizing total correlation; however, this is not an absolute case, even in the case study in this paper. In Figure 7, the red bar on the top right represents Pareto-front given by maximizing both joint entropy and total correlation. If maximizing joint entropy is equivalent to minimizing total correlation, there should be only one optimal solution rather than Pareto-fronts, and its total correlation should be minimum. It seems like there are three (authors') optimal solutions in Figure 7, and it represents the solution which has the maximum joint entropy does not have maximum total correlation, this conflicts with the authors' argument.

3. The original objective function of the MIMR method by Li et al. (2012) is not the one in Function (9) in the paper. To convert the multiobjective problem into a single-objective problem, Li et al. applied the information-redundancy tradeoff weights and maximized the single objective function. In this case, proper weights should be predefined because the optimal solutions can differ due to the weights. If the weight, $\lambda 1$, is equal to one, the problem will become the same with the maxJE what the authors are proposing. To apply the objective functions in Function (9), multiobjective optimization technique should be employed and it will of course yield multiple optimal solutions on Pareto-fronts. In this case, which optimal solution was selected and discussed, such as in Tables 2 and 3, and why the optimal solution was selected should be addressed.

4. Calculating streamflow information for network design from the monthly time series is quite skeptical. Is a hydrometric network which was numerically designed by monthly time series also good for short-term analysis, such as flood forecasting?

5. The authors finally suggested the greedy optimization algorithm. However, the greedy algorithm is not guaranteed to find the global optimum solution and is easy to fall into a local optimum, even though global optimum can be found in the case study of this paper. Also, in the reviewer's opinion, taking 20 years of monthly time series at 12 stations could be not enough to make a general conclusion. Why do we need an optimization technique if we can calculate the objective values of all populations?

---

## Referee Comment (RC3) · Anonymous Referee #3 · 19 Jun 2020

This paper deals with the study of monitoring and design hydrometric networks problems. The authors used information theoretical methods to discuss the objective functions support the choice of a single-objective function to maximize the informative sensor network.

The topic of the paper is very interesting and the problem of finding an optimal monitoring network is certainly a a stimulating challenge. The authors performed an interesting analysis comparing exhaustive optimization and backward greedy approach using

many data but probably I miss the point: why this approach lead to the optimum? And what is the optimal design?

General comments:

1.The paper is mostly well written but there are sometimes redundant informations and figures, tables and formula are presented in an order that confuse the reader. In particular it is convenient that that the explanation of the equation and the symbols involved are immediately after the equation itself. Some figures (for example Fig. 4, 5, 6) need clearer captions and a more detailed description.

2. The authors argue that a single-objective optimization of the joint entropy of all selected sensors will lead to a maximally informative sensor network and that the objective function indirectly minimizes redundant information: in my opinion it is not very clear why this happens. And it seems in contrast with the sentence at line 55 "Minimization of redundancy would mean that each sensor becomes more essential, and therefore the network as a whole more vulnerable to failures in delivering information".

3. The greedy algorithm proposed it is not very clear for me. It is not clear why the optimum found by the algorithm is the global one instead of the local one. Also it is not clear why "remove" a station should be better than a network with a large number of sensors. Probably this is link to other costs (like installation or maintenance costs) but but I missed them if specified in the paper.

4. All the data used in the paper should be used to compare the optimum found by the algorithm with the existent network but they not ensure the optimality.

---

## Referee Comment (RC4) · Anonymous Referee #4 · 19 Jun 2020

Dear Authors,

I've read your manuscript with enthusiasm, as this topic is of my great interest. I do agree with the authors that there is no consensus in the design of monitoring networks, but I do not agree with the conclusions found in this paper. In general, I find that the scoping of the problem validate the obtained results, and therefore, beats its purpose. This is further detailed in the comments below. In addition, I believe that critical methods and literature are overlooked, especially in the use of metaheuristics for the design of sensor networks. Also, I believe that the document could be better structured, as sometimes results and methodology sections overlap.

Objective functions for information-theoretical monitoring network design: what is optimal?

General comments

What is the definition of optimallity?

In the introduction you point out that monitoring objectives are posed as part of a wider decision problem (l18-19). Later, you state that the objective in the design of a sensor network is to maximise its joint entropy (l26-27) as sensor networks may support many decisions. At this point, this becomes a normative approach to the design problem, where you define what optimallity is, and that other objectives should be secondary (l347-348). As a consequence, it is clear that the problem is not a multi-objective optimisation problem anymore, and any trade-off with other objectives (such as minimising redundancy) will directly reflect in a performance loss. This is actually seen in Table 2, where you clearly point it out.

Defining the design problem in these terms makes it sufficiently narrow to justify the use of single-objective optimisation, but the point is that not every problem is.

This also connects with the three main arguments presented in the motivation (s 1.2). l44. "Firstly, we argue that objective functions for optimizing monitoring networks can, in principle, not be justified by case studies" - I do not agree with this postulate. The objective of monitoring is to provide information about the state of a system, to support a given action. Measuring for the sake of measuring do not serve any purpose.

l48. "However, from case studies, we cannot draw any normative conclusions as to what objective function should be preferred." - Preferences are not normative, but relative to the decision problem, objectives and context. These are particular to each case study.

l50. "Secondly, we argue that the joint entropy of all signals together is in principle sufficient to characterize information content and can therefore serve as single optimization

objective" - I do agree with this postulate, as long as the objective is to maximise joint entropy. This goes back to the first comment.

l53. "Thirdly, multi-objective approaches that use some quantification of dependency or redundancy as a secondary objective, next to joint entropy, could only be justified if redundancy is interpreted as beneficial for creating a robust network" - I do partially agree with this postulate, but then again is linked to the definition of optimality. Given the decision problem, a decision-maker may opt to trade some improvement in joint entropy for redundancy (as an example), and that is out of the scope of what this paper presents.

In general, once you assume that joint entropy corresponds to the definition of optimality, the problem is self-validated. Leading to the conclusions that you are presenting such as: "Information theory provides a valid framework for monitoring network design" (l344), and that single-objective optimisation is sufficient to approach the sensor network design problem.

What are the alternatives for solving the optimisation problem?

From the methodological point of view, I see that you propose the greedy drop algorithm, as an alternative to greedy selection, especially attractive in the cases where exhaustive search is not feasible (l216-223). However, you omitted mentioning metaheuristics to approach this issue. These are often used in the design of monitoring networks when the combinatorial problems are too large. This being the argument for greedy approaches.

What are the cases of sensor network design that are considered?

When designing a monitoring network, you may have one of 3 possible scenarios: augmentation, reduction or relocation. Augmentation makes for the case when additional sensors are to be placed in the network. Reduction, accounts for the case when sensors would like to be removed from the network. Relocation, deals with the issue of

changing the placement of the sensors. The case you illustrate in this paper corresponds to reduction, as the objective is to select a sub-set of sensors from a larger pool. However, you use justifications from the augmentation case (l221-222) to justify the use of a greedy algorithm. I think it is necessary to better define the optimisation problem in this respect.

Specific comments

The section on presenting the basics of information theory can be better summarised. There is plenty of "well-known" material on it.

l172-173 requires a reference.

l187 requires explanation about what is objective GR3

l200-201 These are not MOO methods. These are objective functions.

l202- 203 requires Preferences

l218-220 It is not true that the only way of selecting stations is using greedy algorithms.

l221 is not combinatorial "explosion". Instead we can argue that the problem is exponentially complex ($O^n$)

l222 I think here you are mixing two design problems. One of the problems is of design (where to measure at several locations), and other of augmentation (Where to put additional stations). Of course these are clearly different processes.

l232 I think it should be necessary to include methods using metaheuristics for comparison.

l233 It is not clear what logistical reasons are. Should not these be included in the optimisation constraints?

l235-236 This seems speculative at this point, and better be moved to other place in the document (perhaps introduction?)

l233 This "golden standard" expression seem somewhat loose talk. Can just point out is the only way to prove optimallity?

Figure 4 can be improved. labels are hard to read, and would be more informative just to keep the ID's?. Also if its for monthly data, have you consider using flow duration curves instead? alternatively, please consider using a log y-scale, as discharge distributions are positively skewed. In addition, y-scale label is missing.

in Table 2 are presented the results of different optimisation criteria (defined as MOO), but there is no indication of the selecting strategies, thus the information of the whole Pareto set is unavailable. In top, that is the whole point of MOO, that there exists trade-offs between objectives that cannot be assessed by the modeller.

S4.2 includes parts that should have been presented in the methodology and not in the results section.

l276. H(X,X) = H(X), therefore two completely "dependent" are exactly as informative as one.

Visualisation using Venn diagrams are an excellent way of presenting concepts, but are really hard to follow to describe precise quantities. Will it be possible to re-think Figure 5 in a simpler manner?

l287-288 Those are precisely the trade-offs that decision-makers do in MOO, and the reason of its relevance. If you claim that maximum joint entropy is somewhat equivalent to minimum total correlation, then these two objective functions are not conflicting, and therefore, by maximising one, you are maximising the other. Precisely as shown in Figure 7.

l297-298 The whole point in obtaining the Pareto front between maximum joint entropy and minimum total correlation explores the trade-offs between a network that is able to capture most information, vs a network that has little "information overlap", which is not the same as the individual entropies are different. Therefore, these are different

metrics, and is the reasoning behind finding the Pareto set in the dashed line of Fig 7.

S4.3 A lot of the text in this section can be part of the methodology.

Table 4 is quite hard to read. Also, this table is precisely showing you that optimallity is not found using greedy algorithms.

Table 5 Be consistent in the amount of significant digits through your document

l344 No information system is justified if there is not objective to tackle. What problem am I addressing if I do not know what the problem is.

l361 Large optimisation problems are tackled using metaheuristics, and has been a widely used approach. This has not been mentioned here at all.

l364 the differences between the greedy and exhaustive search approaches have not been presented quantitatively. In this problem they may seem "little" (not being explicit about what little or big is in the context of the problem), but this cannot be ensured for larger problems that the ones presented here.

l367-368 Language has to be precise (how to numerically calculate this objective function, or other objective functions used in other approaches)

l370 Any information metric is hard to calculate with limited data.

l378-379 I completely agree with this line "before thinking about how to optimize, we should be clear on what to optimize".

l381 I visited the GitHub repository, but I was unable to find the code to reproduce these results. Only a reply to a WRR paper of 2018, and a fork of pysheds.

---

## Editor Comment (EC1) · Nunzio Romano (Editor) · 19 Jun 2020

Dear Authors, Your original submission received interesting comments from the selected reviewers. There still is some time before the closure of the discussion, and I am wondering whether you can post some short and very direct replies on key comments, so as to stimulate perhaps some additional and useful comments from our reviewers.

---

## Author Comment (AC1) · 27 Jul 2020

Dear Editor,

We are grateful for being blessed with four reviewers providing detailed comments and suggestions. We have included extensive point by point replies to all four reviewers. We also have several concrete ideas how to benefit from the reviews to improve the clarity of the paper.

We noticed that some of the reviewers were not yet convinced by our key point about minimizing redundancy not being needed as an additional objective in monitoring net-

work optimization. Perhaps this is not surprising as this goes against ideas brought forward in many published papers, and is therefore necessarily controversial.

As you can see for example from our replies to the most extensive comments and disagreement by reviewer 4, we firmly stand by our arguments and perhaps explain them more clearly in the replies.

We also want to clarify here that our main point is not that joint entropy is the only objective needed in any situation, which is an impression that may have risen with some of the reviewers. We merely argue that the only reason why high redundancy is not desirable, is the information loss it causes, which is already reflected in the maximization of joint entropy.

We hope that our extensive replies to the comments will perhaps convince the reviewers, and that incorporating the many useful comments will make our manuscript clearer.

Best regards, Hossein Foroozand and Steven Weijs

---

## Author Response (AR1)

**Combine answers to all 4 reviewers**

**Reply to comments of reviewer #1**
(reviewer comments in black, replies in blue)

This paper deals with the optimisation of monitoring networks using information-theoretical methods with a focus on the analysis of useful objective functions. The authors argue that a single-objective optimization of the joint entropy of all selected sensors will lead to a maximally informative sensor network. They compare exhaustive optimization, a greedy approach and a new "greedy drop" approach using available monthly runoff data.

I enjoyed reading the manuscript, especially the introduction to the information theory terms. This study adds some interesting new views on the optimisation of monitoring networks and fits well to the scope of HESS. However, although the paper is mostly well written, it is sometimes poorly structured. In addition, I have some general and specific remarks that needs to be considered before publication. I therefore recommend a minor revision of the paper.

We thank Dr. Heye Bogena for his positive review and constructive suggestions, which allowed us to improve the quality of the manuscript. In the following section, reviewer comments are in black color, author responses are in blue color, and changes proposed for the text in the manuscript are in *italics* and underline font.

General comments: For this study, monthly runoff data were used from gauging stations with very different catchment areas. It can be assumed that the gauging stations with smaller catchment areas will show a higher temporal variability of discharge. Therefore, their sub-monthly data should have a higher information value than the data of the stations with larger catchments. Thus, the authors should also discuss the dependence of their results on the temporal scale of the discharge data used.

We agree with the reviewer's comment on the connection of temporal scale and having a higher information value. Increasing resolution on a temporal scale means having more data points, which in turn, may translate to having higher information value and better estimation of network dependencies. However, in our case, the low entropy value of gauging stations with smaller catchment areas is the outcome of having coarse discretization rather than the effect of temporal scale. In fact, if we use finer discretization, we will get higher entropy values for gauging stations with smaller catchment areas even with a monthly temporal scale. As we outlined in section 3, our goal in this paper was to control for the effect of temporal variability of data and quantization method in our methodology comparison. To do this, we used the same data period, with the same temporal scale and quantization method proposed by Li et al. (2012). Although our paper focuses on the role of the objective function in the network design, we think it is helpful for the readers to highlight the effect of decisions we made to isolate the impact of temporal variability of data and quantization method. Therefore, we expanded the first paragraph of the further work section to discuss these issues.

Added in L240:

In this paper, we focused on the theoretical arguments for choosing the right objective functions to optimize, and compared a maximization of joint entropy to other methods, while using the same data set and quantization scheme. Another important question that needs to be addressed in future research is how to numerically calculate this objective function, or other objective functions used in other approaches. What many of these objective functions have in common, is that they rely on multi-variate probability distributions. For example, in our case study, the joint entropy is calculated from a 12-dimensional probability distribution. These probability distributions are hard to reliably estimate from limited data. *Also, these probability distributions are influenced by multiple factors, including data's temporal scale and quantization assumptions. To have an unbiased comparison framework of objective functions, we chose to isolate the effect of temporal variability of data and quantization method on methodology comparison. It is worthy to acknowledge that these assumptions as well as data availability can greatly influence optimal network ranking, and require more attention in future research.*

The order of presentation of the tables and figures is sometimes confused.

Thanks for pointing this out. The confusion is caused by the placing of table1 at the end of the manuscript. This movement is forced by the Copernicus Latex standard for the discussion manuscript format (partly because it takes a whole page ). We'll make sure all tables are in a proper position in the final publication format.

Specific comments:
 L1: "layout"

Thanks for catching this, we will correct it.

L11-83: Subtitles within the introductory chapter are rather uncommon.
Thank you for your comments. We now removed all subtitles and modified the introduction section, and moved down the scope subsection under the methodology section.

L123-145: Please always give the equations directly after their first citation.
Will be fixed as recommended.

L166: You are first referring to Appendix B instead of A. Should be reversed.
Will be fixed as recommended.

L184: Which part of the Appendix?
Thanks, it is properly referenced now.

L243: Eq. 12 and the corresponding variable description should directly follow this sentence.
Will be fixed as recommended.
L259: "Tables"
Thanks for catching this, it is fixed.

L259-260: Please indicate the location of the eight stations containing the joint information of all 12 stations in Figure 3 and discuss whether the result is meaningful, e.g. in terms of the distance between stations, their respective catchment areas, etc..

Thanks for your comment. We add a new figure that contains the information of the top eight stations of all four optimization methods. Although information theory-based methods do not consider the distance between stations as a factor in network design, there are noticeable differences in terms of the distance between selected stations by different methods. In addition to the new figure, we will expand our discussion on this.

Added in L269:

To assess and illustrate the workings of the different objectives in retrieving information from the water system, we compared three existing methods with a direct maximization of the joint entropy of selected sensors, H (S,Fc), indicated with maxJE in the results, such as Tables 2 and 3. The joint entropy results in table 2 indicate that maxJE is able to find a combination of 8 stations that contains joint information of all 12 stations ranked by other existing methods. *Figure 5 displays spatial distribution of the top 8 stations chosen by different methods. Before any interpretation of the placement, we must note that the choices made in quantization and the availability of data play an important role in the optimal networks identified. Whether the saturation that occurs with 8 stations has meaning for the real world case study depends on whether the joint probability distribution can be reliably estimated. This is highly debatable and merits a separate detailed discussion which is out of the scope of this paper. We present this case study solely to illustrate behaviour of the various objectives.*
*The most notable difference between MaxJE and the other methods is the selection of all 3 of the stations located most downstream. While other methods would not select these together due to high redundancy between them, maxJE still selects all stations, because despite the redundancy, there is still found to be enough new information in the second most downstream station. This can be in part attributed to the quantization choice of equally sized bins throughout the network, leading to higher information contents downstream. While this quantization choice is debatable, it is important, in our opinion, to not compensate artifacts from quantization by modifying the objective function, even if the resulting network may seem more reasonable, but rather to address those artifacts in the quantization choices themselves. To repeat the key point: An objective function should not be chosen based on whether it yields a "reasonable network" but rather based on whether the principles that define it are reasonable.*

L260-265: Results shown in Figs. 5 and 6 need to be explained in more detail.
Agreed, we expanded our discussion on these two figures (after adding new figure they become fig 6 and 7) by adding the following changes:

We propose to modify as follows:

We demonstrate that other methods with a separate minimum redundancy objective lead to the selection of stations with lower new information content (green area in Figure 6). This leads to slower reduction of the remaining uncertainty that could be resolved with the full network. *Reduction of the yellow area in each iteration (i.e. the information loss compared to the full network) in Figure 6 corresponds to the growth of joint entropy values in Table 2 for each method. maxJE has the fastest, and minT the slowest rate of reduction of information loss. A method's preference for reaching minimum redundancy or growing joint information (red area in Figure 6) governs the reduction rate of information loss.* Also, Figure 7 provides auxiliary information about the evolution of pairwise information interaction between already selected stations $<X_1, X_2,...,X_{i-1}>$ in the previous iterations and new proposed station $X_i$. Figure 7 illustrates the contrast between the choice of the proposed stations in the first six iterations by different methods. *For instance, minT method aims to find a station that has minimum mutual information (red links in Figure 7) with already selected stations. In contrast, the maxJE method tries to grow joint entropy, which translates to finding a station that has maximum conditional entropy (green segments in Figure 7). Other methods opt to combine two approaches by either imposing a constraint (WMP) or having a trade-off between them (MIMR).*

L305-309: Repetition

Thanks for pointing this out. We now modified the paragraph to solve the issue:

*Different search strategies have been adopted in the literature for monitoring network design. The most commonly used greedy algorithms impose a constraint on exhaustive search space to reduce computational effort. We investigated three different search strategies to obtain the optimal network in the context of using maxJE as an objective function. We discuss the advantages and limitations of each search strategy in terms of optimality of the solution and computational effort.*

L315-319: This estimate is of limited value because it depends largely on the programming code (i.e. Matlab is very slow compared to e.g. FORTRAN).

Agreed, Matlab may not be the most time-efficient programming code. But, our main message was to highlight the exponential explosion in the number of combinations as the number of stations increases linearly. So we can see the two numbers relative to each other. We will clarify in the text the the relative differences and scaling are more important than the absolute numbers. These are language and implementation independent. :.

*The computational burden is therefore greatest when about half of the stations are selected. For a number of potential sensors under 20, this is still quite tractable (4 minutes on normal PC, implemented -by a hydrologist- in MATLAB, with room for improvement by optimizing code, language, and programmer), but for larger numbers, the computation time increases very rapidly. When considering all sub-network sizes, the number of combinations to consider is 2^n, so an exponential growth. We could make an optimistic estimate, only considering the scaling from station combinations to evaluate, but not considering the dimensionality of the information measures. For 40 stations, this estimate would yield a calculation time of more than 5 years, unless a more efficient 1 algorithm can be found. Regardless of potential*

*improvements in implementation, the exponential scaling will cause problems for larger systems.*

L323: "Tables"
Thanks for catching this, it is fixed.

L325: Table 6 deserves more explanation. The numbering of the tables is confusing. This should be rather Table 4.
Thanks for catching this, the numbering issue is fixed (it becomes Table 4). Also, we expand our discussion:

We will add in L360:

*Results in Table 4 show multiple network layouts with equal network size and joint information exist. For this case, network robustness could be an argument to prefer the network with maximum redundancy. Also, it should be acknowledged that the assumptions in data quantization would influence reaching equal joint information, and further research is warranted to investigate the network's susceptibility to quantization assumptions.*

L328-329: If find Table 4 difficult to understand. In addition, the captions of Table 4 and 5 indicate 240 data points, which should be rather 860*12 data points, if I understand correctly. I suggest combining Table 4 and 5.

Thanks for catching this (it was a mistake in our first submission before the start of the online discussion). it is already corrected in the downloadable pdf file. We artificially generated 860 data points per station based on statistics from the original data (240 data points).

L339: "generated"
Will be fixed as recommended.

L344-356: This section is more like introduction and discussion and thus not appropriate for a concluding chapter. L357-363: You must present the most important results of your study more clearly, e.g. by using also bullet points.

Thanks for pointing this out. We will thoroughly revise both structure and language of this section (L344-363) to be more direct and clear. The following is the modified version:
Added in L387:

*The aim of this paper was to contribute to better understanding the problem of optimal monitoring network layout using information-theoretical methods. Since using resulting networks and performance metrics from case studies to demonstrate that one objective should be preferred over the other would be circular, the results from our case study served as an illustration of the effects, but not as arguments supporting the conclusions we draw about objective functions. We investigated the rationale for using various multiple-objective and single-objective approaches, and discussed the advantages and limitations of using*

*exhaustive vs. greedy search. The main conclusions for the study can be summarized as follows:*

- *The purpose of the monitoring network governs which objective functions should be considered. When no explicit information about users and their decision problems can be identified, maximizing the total information collected by the 395 network becomes a reasonable objective. Joint entropy is the only objective needed to maximize retrieved information, assuming that this joint entropy can be properly quantified.*
- *We argued that the widespread notion of minimizing redundancy, or dependence between monitored signals, as a secondary objective is not desirable and has no intrinsic justification. The negative effect of redundancy on total collected information is already accounted for in joint entropy, which measures total information net of any redundancies.*
- *When the negative effect on total information is already accounted for, redundant information is arguably beneficial, as it increases robustness of the network information delivery when individual sensors may fail. Maximizing redundancy as an objective secondary to maximizing joint entropy could therefore be argued for, and trade-off between these objectives could be explored depending on the specific case.*
- *The comparison of exhaustive and greedy search approaches shows that no greedy approach can exist that is guaranteed to give the true optimum subset of sensors for each network size. However, the exponential computational complexity, which doubles the number of sensor combinations to evaluate with every sensor added, makes exhaustive search prohibitive when the number of possible locations become larger than about 25. The complexity of the greedy approach is quadratic in the number of locations, and therefore feasible for large search spaces.*
- *The constraints to the search space imposed by the greedy approach could also be interpreted as a logistical constraint. In a network expansion scenario, it disallows replacement of stations already selected in the previous iteration.*
- *We introduced the "greedy drop" approach that starts from the full set and deselects stations one by one. We have demonstrated that the two types of greedy approaches do not always lead to the same result, and neither approach guarantees the unconstrained true optimal solution. Synergistic interactions between variables may play a role, although this is not the only possible explanation. In our case study, the suboptimality of greedy algorithms was not visible in original data, but we demonstrated its existence with artificially generated data. In our specific case studies, differences between exhaustive and greedy approaches were small; especially when using a combination of the greedy add and greedy drop strategy. It remains to be demonstrated in further research how serious this loss of optimality is in a range of practical situations, and how results compare to intermediate computational complexity approaches such as metaheuristic algorithms.*

L365-379: After removing any redundancies this section should be moved to the discussion section.

We modified Further work section by addressing reivewer's first comment on the need to discuss the dependence of our results on the temporal scale of the data. In an effort to bring the important points about objective function across, we left other issues undiscussed. We think having a separate section to give direction to future research would be helpful for our readers.

**Reply to comments of reviewer #2**

(reviewer comments in black, replies in blue)

We are glad that reviewer #2 was interested in reading our paper. We thank reviewer #2 for giving an example that our argument may become agreeable while raising theoretical questions about some hypothetical situations that are not discussed in our paper. Although some of the pointed questions are out of the scope of this paper, we think addressing these concerns would clarify our underlying assumptions and help our readers. In the following section, reviewer comments are in black color, author responses are in blue color, and changes planned for the text in the manuscript are in *italics* and underline font.

This paper discussed the objective functions in information theory-based hydrometric network design problems and suggested a backward greedy approach instead of other optimization methods. While I was interested in reading this paper because taking more correct, reasonable, and meaningful objective functions and proper optimization techniques is very important in the network design using information theory, unfortunately, I couldn't get the answer of the question in the title, "what is optimal?", throughout this paper.

The rhetorical question "what is optimal?" in the title can be understood in two ways. Firstly it may refer to the question "what is the optimum network configuration?" (given that we have defined what we want from the network, i.e. the objective function). Secondly it might ask the question "what do we want from the network?"; "what do we consider optimal?"
Putting this question in the title was our way of drawing the reader's attention to the core difference between these two. Also it serves to hint at the fact that choice of objective function controls optimal answer—changing the objective function would change the subsequent optimal answer, so the objective function must be justified by reasoning independent of the numerical results. We will add a brief explanation in the beginning of the introduction.

Added in L23:
*In this paper, we do not answer the question "what is optimal?" with an optimal network. Rather, we reflect on the question of how to define optimality in a way that is logically consistent and useful within the monitoring network optimization context, thereby questioning the widespread use of minimum dependence between stations as part of the objectives.*

1. The authors argue that a higher amount of redundant information should be preferred because it reinforces the robustness of the network. However, this is not a general statement but can be applied only for a specific condition.

In the abstract section, we stated that '' for two networks of equal joint entropy, one with a higher amount of redundant information should be preferred for reasons of robustness against failure.'' We did not intend to make a general statement. We made this conditional statement about two networks of equal joint entropy to point out the fact that minimizing dependence as a secondary objective would give preference to one with lower shared information. We think this is undesirable in any kind of situation. Our main point of the paper is that there is, in general, no reason for minimizing redundancy that is not already reflected in the max joint entropy objective. We emphasize this point by saying that we may even prefer maximizing instead of minimizing redundancy.

World Meteorological Organization has recommended the minimum network densities, which do not represent the optimal number of stations, rather they are suggested to avoid any serious deficiency in water resources management. That is, just meeting the guideline couldn't be sufficient while there is no clear definition of sufficient density. However, many regions in the world, such as developing countries, do not meet even the WMO guideline. Even in the study area in this paper, Brazos River Watershed, the drainage area is about 116,000 sq.km. and there are 12 USGS stations considered as the existing stations. The current network density becomes 9,667 sq.km per station while the minimum network density of the WMO guideline for the Interior Plains is 1,875 sq.km., which is more than five times sparser network. Considering the WMO's minimum network is the baseline not to lose critical information, it should be noted that the network is seriously under-gauged, such that we need to consider network expansion and network efficiency rather than network robustness. In this case, minimizing total correlation is more meaningful to optimize network efficiency. Besides, installing and maintaining monitoring stations often highly depend on financial budgets which cannot be satisfactory in practice. On the other hand, if the network density largely exceeds the minimum network density, and the water resources managers consider shrinking network by closing stations, the authors' argument may become agreeable.

Thanks for your comment. In general, network density is an important issue, and we acknowledge that this network is under-gauged. However, decision on network density is mainly derived by budget and network's purpose, and formulating objective function for any given budget is discussed in this paper. In fact, our main focus is to discuss the formulation of information-theoretical objective functions and previous literature on that topic. In monitoring network design problems, there are multiple important contributing factors, including assumptions on network density, objective function, data quantization, temporal variability of data, etc. We restricted our scope to avoid multifarious analysis and isolate the effect of all contributing factors except assumption on the objective function. Therefore, we made several edits to section 3 and 4.1 to clarify these contributing factors to our readers.

Some further related thoughts, that we integrated at several points in the text:
- We agree that network robustness it's the secondary objective that only becomes relevant when the first objective is met, and is less important in under gauged situations.
- We do not propose a multi objective approach here, but added this only to emphasize that minimizing redundancy is not a helpful objective.
- The first objective of maximizing information content per sensor, or network efficiency, is optimized by joint entropy. Adding other objectives in a multi-objective optimization will detract from that primary objective.
- The question of how to properly define network efficiency is independent from whether the current network is under gauged or over gauged or whether we are considering expanding or reducing the network.

Changed start of section 3::
*In previous studies, the focus of the research has been on finding an optimal network for the subject*
*case study without sufficiently addressing the theoretical justification of applying a new methodology.*

*For this reason and the primary goal of this paper, which is highlighting the unnecessary use of*

*multiple objective functions in monitoring network design, we decided to apply our methodology in*

*Brazos River streamflow network (Figure 3) since this network was subject of study for the MIMR*

*method. This network is under-gauged, according to the World Meteorological Organization density*

*requirement. However, it provides an opportunity to eliminate subjectivity issues on network design*

*decisions such as network density, data quantization, and temporal variability of data.*

We also added to section 4.1:
*Whether the saturation that occurs with 8 stations has meaning for the real world case study depends on whether the joint probability distribution can be reliably estimated. This is highly debatable and merits a separate detailed discussion which is out of the scope of this paper. We present this case study solely to illustrate behaviour of the various objectives.*

2. The authors also argue that maximizing joint entropy already connotes minimizing total correlation; however, this is not an absolute case, even in the case study in this paper. In Figure 7, the red bar on the top right represents Pareto-front given by maximizing both joint entropy and total correlation. If maximizing joint entropy is equivalent to minimizing total correlation, there should be only one optimal solution rather than Pareto-fronts, and its total correlation should be minimum. It seems like there are three (authors') optimal solutions in Figure 7, and it represents the solution which has the maximum joint entropy does not have maximum total correlation, this conflicts with the authors' argument.
We agree that this is not the most accurate formulation and we will clarify.

We do not intend to claim that minimizing redundancy is equivalent to maximizing joint entropy. Rather, the effects of redundancy on the amount of information captured, which we could see as some form of inefficiency, is already captured in the joint entropy measure, which quantifies total information net of any redundancies.
Because also the marginal entropy of the stations plays a role in joint entropy, there is still a trade-off, but this is not a trade-off that represents pareto-optimality in any sensible way: the only reason to consider minimizing total correlation is already captured fully in the competing objective, joint entropy.
We modified our text to make it clear that it is not equivalent, but the reason to include minimum redundancy is already covered by maximizing joint entropy.

As mentioned in caption in figure 7, past approaches, in the literature, gave importance to the maximum joint entropy and minimum total correlation. We make a conditional statement that if a trade-off between objectives is to be considered, the red Pareto-front should be preferred instead of the black line. Also, before making this conditional statement, we made a general statement that tradeoff is irrelevant in our paper and information can be maximized with the

horizontal direction only. Therefore, the conditional statement is not in conflict with our main argument since our method does not consider a tradeoff and will only produce one optimal solution. We modified the figure's caption to clarify this issue.

 Added in figure's caption:

The resulting total correlation and joint entropy for all 924 possible combinations of 6 out of 12 sensor locations. In some past approaches, a pareto front in the lower right corner is given importance. In the paper, we argue that this trade-off is irrelevant, and information can be maximized with the horizontal direction only. If a trade-off with reliability _needs to be_ considered, the Pareto front of interest is in the top-right corner _instead of the lower right corner that is previously recommended in the literature._

3. The original objective function of the MIMR method by Li et al. (2012) is not the one in Function (9) in the paper. To convert the multiobjective problem into a singleobjective problem, Li et al. applied the information-redundancy tradeoff weights and maximized the single objective function. In this case, proper weights should be predefined because the optimal solutions can differ due to the weights. If the weight, λ1, is equal to one, the problem will become the same with the maxJE what the authors are proposing. To apply the objective functions in Function (9), multiobjective optimization technique should be employed and it will of course yield multiple optimal solutions on Pareto-fronts. In this case, which optimal solution was selected and discussed, such as in Tables 2 and 3, and why the optimal solution was selected should be addressed.

Our responses to this comment are three-fold:

1)    Function (9) in our paper is the same as Equation 14 in Li et al. (2012). We directed our reader to Table1 for more details where tradeoff weights are explained.

2)     Li et al. (2012) stated that "_the sensitivity analysis was only carried out for information weight falling between 0.5 and 1.0. Results are summarized in Table 4, which mainly signify the stability of MIMR with respect to information weight._". Therefore, we chose to accept Li et al. (2012)'s conclusion on insignificant effect of information-redundancy tradeoff weights in this particular case study. MIMR's information presented in our Tables 2 and 3 can be verified by the information presented in Li et al. (2012)'s Table 3.

3)    We disagree with the reviewer's statement: "_If the weight, λ1, is equal to one, the problem will become the same with the maxJE what the authors are proposing._". Li et al. (2012) proposed λ1 as trade-ff weight for both maximum joint entropy (H) and maximum transinformation (T). Therefore, if λ1 is equal to one, MIMR method would have two terms (H and T) in its objective function while maxJE method has only one term (H). In any circumstance, MIMR will not become the same as the maxJE method.

4. Calculating streamflow information for network design from the monthly time series is quite skeptical. Is a hydrometric network which was numerically designed by monthly time series also good for short-term analysis, such as flood forecasting?

Hydrometric network design with the purpose of flood forecasting not only requires data set with high temporal resolution, it may also need a combination of rain gauges and streamflow gauges. As we outlined in section 3, our goal was to isolate the effect of temporal variability of data and quantization method from the methodology comparison; we used the same data period with the same temporal scale and quantization method proposed by Li et al. (2012). Although we intended to only focus on the role of the objective function in the network design, we think it is helpful for the readers to highlight the effect of decisions we made to isolate the impact of temporal variability of data and quantization method. Therefore, we will expand the first paragraph of the further work section to address these issues.

Added in L240:

In this paper, we focused on the theoretical arguments for choosing the right objective functions to optimize, and compared a maximization of joint entropy to other methods, while using the same data set and quantization scheme. Another important question that needs to be addressed in future research is how to numerically calculate this objective function, or other objective functions used in other approaches. What many of these objective functions have in common, is that they rely on multi-variate probability distributions. For example, in our case study, the joint entropy is calculated from a 12-dimensional probability distribution. These probability distributions are hard to reliably estimate from limited data. *Also, these probability distributions are influenced by multiple factors, including data's temporal scale and quantization assumptions. To have an unbiased comparison framework of objective functions, we chose to isolate the effect of temporal variability of data and quantization method on methodology comparison. It is worthy to acknowledge that these assumptions as well as data availability can greatly influence optimal network ranking, and require more attention in future research.*

5. The authors finally suggested the greedy optimization algorithm. However, the greedy algorithm is not guaranteed to find the global optimum solution and is easy to fall into a local optimum, even though global optimum can be found in the case study of this paper. Also, in the reviewer's opinion, taking 20 years of monthly time series at 12 stations could be not enough to make a general conclusion. Why do we need an optimization technique if we can calculate the objective values of all populations?

We agree with the reviewer's comment on greedy algorithms. In section 4.3, we reiterated that only exhaustive optimization will give the true optimum, which is global optimum solution. We stated (Line 319) most of the previous approaches listed in Table 1 can be categorized as greedy optimizations. The comparison between greedy and exhaustive optimization has two benefits: (1) it raises awareness in the scientific community that greedy optimization might fall into local optimum; (2) it shows why exhaustive optimization is not feasible in higher dimensions.

Our response to reviewer's opinions is as follows:

Regarding the first opinion "taking 20 years of monthly time series at 12 stations could be not enough to make a general conclusion.", we explained our reasons for choosing this data set in section 3 as well as our answer to comment #1. It is important to emphasize that unbiased comparison between objective functions is the goal of our paper. Therefore, we decided to use a data set that was subject of study for the MIMR method (a highly cited paper in our field). We agree that data scarcity is an issue in drawing conclusions on the case study, be the objective of the case study is just for illustration of the discussion on objective functions.

Regarding the second opinion: we agree we can just use exhaustive optimization here, where all objective function values of the search space are calculated. However, due to the exponential growth in search space, this is practically impossible in other cases. That is why we discuss other, more tractable approaches.

**Reply to comments of reviewer #3**

(reviewer comments in black, replies in blue)

This paper deals with the study of monitoring and design hydrometric networks problems. The authors used information theoretical methods to discuss the objective functions support the choice of a single-objective function to maximize the informative sensor network. The topic of the paper is very interesting and the problem of finding an optimal monitoring network is certainly a stimulating challenge. The authors performed an interesting analysis comparing exhaustive optimization and backward greedy approach using many data but probably I miss the point: why this approach lead to the optimum? And what is the optimal design?

We are glad that reviewer #3 has found the topic of our paper very interesting. We really appreciate your thoughtful comments that have guided us to improve the manuscript. In the following section, reviewer comments are in black color, author responses are in blue color, and changes made to the text in the manuscript are in *italics* and underline font.

In our paper, true optimum refers to optimal network ranking, which is influenced by the choice of objective function. Our goal is to advocate for choosing objective function based on theoretical argument. In this context, we presented three main arguments, which subsequently resulted in proposing maxJE method. We argued notions of minimizing dependence or redundant information by a secondary objective function between monitored signals are not desirable and have no intrinsic justification.

The greedy approach does not give the true optimum, but is computationally more tractable for larger problems. Based on suggestions of reviewer 4, we also will mention some other approached of intermediate complexity.

We added in L372:

*Based on our limited case study, the questions remain open: 1) whether faster algorithms can be formulated that yield guaranteed optimal solutions, and 2) in which cases the greedy*

*algorithm provides a close approximation. It is also possible to formulate modified greedy methods with the ability of replacing a limited number of monitors instead of just adding monitors. This leads to a significantly reduced computational burden compared to exhaustive optimization, while reaching the optimum more often than when adding monitors one at a time. In Table 5, it can be seen that allowing a maximum number of two relocated monitors would already reach the optimal configurations for this specific case. Another limitation of this comparison is that we did not consider metaheuristic search approaches (Deb et al., 2002; Kollat et al., 2008), which fall in between greedy and exhaustive approaches in terms of computational complexity, could serve to further explore the optimality versus computational complexity trade-off. It would be interesting to further investigate what properties in the data drive the suboptimality of greedy algorithms. Synergistic interactions (Goodwell and Kumar, 2017) are a possible explanation, although our generated data example shows that even when moving from 1 to 2 selected stations, a replacement occurs. Since there are only pairs of variables involved, synergy is not needed in the explanation of this behaviour. Rather, the pair with maximum joint entropy does not always include the station with maximum entropy, which could perhaps be too highly correlated with other high entropy variables.*

General comments:

1.The paper is mostly well written but there are sometimes redundant informations and figures, tables and formula are presented in an order that confuse the reader. In particular it is convenient that the explanation of the equation and the symbols involved are immediately after the equation itself. Some figures (for example Fig. 4, 5, 6) need clearer captions and a more detailed description.

Thanks for the evaluation of our paper as mostly well written. We have made significant changes to the manuscript in response to your suggestions. The changes are summarised as follows:

- We now provide explanation of the equation and the symbols directly after the first appearance.
- Regarding the confusion in table orders, it was caused by the placing of table1 at the end of the manuscript. This movement is forced by the Copernicus Latex standard for the discussion manuscript format (partly because it takes a whole page ). We'll make sure all tables are in a proper position in the final publication format.
- We modified fig 4 caption and provided more information on how the presented information is calculated.

Planned additions to figure 4 caption:

Brazos River streamflow (m3/s) statistics and resulting entropy values (bits). *The stations' IDs are organized from upstream to downstream gauges in the watershed. Entropy values are calculated by floor function and parameter a=150 m3/s.*

- We expanded our discussion on fig 5 and 6 (after adding new figure they become fig 6 and 7) by adding the following changes:

Added in L287

*Though already necessarily true from the formulation of the objective functions, we use the case study to illustrate how other methods with a separate minimum redundancy objective lead to the selection of stations with lower new information content (green area in Figure 6). Reduction of the yellow area in each iteration (i.e. the information loss compared to the full network) in Figure 6 corresponds to the growth of joint entropy values in Table 2 for each method. maxJE (by definition) has the fastest, and minT the slowest rate of reduction of information loss. Methods' preference for reaching minimum redundancy or growing joint information (red area in Figure 6) governs the reduction rate of information loss. Also, Figure 7 provides auxiliary information about the evolution of pairwise information interaction between already selected stations $<X_1, X_2, ..., X_{i-1}>$ in the previous iterations and new proposed station $X_i$. Figure 7 illustrates the contrast between the choice of the proposed stations in the first six iterations by different methods. For instance, minT method aims to find a station that has minimum mutual information (red links in Figure 7) with already selected stations. In contrast, the maxJE method tries to grow joint entropy, which translates to finding a station that has maximum conditional entropy (green segments in Figure 7). Other methods opt to combine two approaches by either imposing a constraint (WMP) or having a trade-off between them (MIMR).*

2. The authors argue that a single-objective optimization of the joint entropy of all selected sensors will lead to a maximally informative sensor network and that the objective function indirectly minimizes redundant information: in my opinion it is not very clear why this happens. And it seems in contrast with the sentence at line 55 "Minimization of redundancy would mean that each sensor becomes more essential, and therefore the network as a whole more vulnerable to failures in delivering information".
We agree that this is not the most accurate formulation and we will clarify.

We do not intend to claim that minimizing redundancy is equivalent to maximizing joint entropy. Rather, the effects of redundancy on the amount of information captured, which we could see as some form of inefficiency, is already captured in the joint entropy measure, which quantifies total information net of any redundancies.
Because also the marginal entropy of the stations plays a role in joint entropy, there is still a trade-off, but this is not a trade-off that represents pareto-optimality in any sensible way: the only reason to consider minimizing total correlation is already captured fully in the competing objective, joint entropy.
On the other hand, putting an extra focus on minimizing redundancy by adding it as an extra objective, will focus on independence (and thus on how essential the sensors are, increasing vulnerability to failure), without caring for whether it makes a positive contribution to the total non-redundant information collected (which in our opinion is the only motivation for decreasing dependence).

We will modify our text to make it clear that it is not equivalent, but the reason to include minimum redundancy is already covered by maximizing joint entropy.

Added in L56:

Secondly, we argue that, in purely information-based approaches, the joint entropy of all signals together is in principle sufficient to characterize information content and can therefore serve as single optimization objective. *Notions of minimizing dependence between monitored signals through incorporation of other information metrics in the objective function lack justification and are therefore not desirable. Thirdly, because the undesirable information inefficiencies associated with high dependency or redundancy are already accounted for in maximizing joint entropy, we could actually argue for maximizing redundancy as a secondary objective, because of its associated benefits for creating a network robust against failures of individual sensors. Minimization of redundancy would mean that each sensor becomes more essential, and therefore the network as a whole more vulnerable to failures in delivering information. Adding a trade-off with maximum redundancy is outside the scope of this paper, but serves to further illustrate the argument against use of minimum redundancy.*

3. The greedy algorithm proposed it is not very clear for me. It is not clear why the optimum found by the algorithm is the global one instead of the local one. Also it is not clear why "remove" a station should be better than a network with a large number of sensors. Probably this is link to other costs (like installation or maintenance costs) but I missed them if specified in the paper.

The main goal of this paper is to compare and argue about choice of objective function in the information-theoretical monitoring network. We found that most of the previous approaches (listed in Table 1) used greedy optimization as a tool for their respective objective functions. Therefore, we decided to study greedy and exhaustive optimization to raise awareness in the scientific community that greedy optimization might fall into local optimum. The results show that although the greedy algorithm found a global optimum in our case study, it only found a local optimum in the added case with a artificial data set. As you correctly state, it is not generally true that greedy algorithms will find the global optimum.

The "removal" of a station is not a literal removal, it is meant to indicate ranking stations starting from the full set as opposed to starting from the empty set. We indicated that both greedy add and drop might not produce true optimum, and only exhaustive optimization will give the true optimum since it does not impose a constraint on search space.

4. All the data used in the paper should be used to compare the optimum found by the algorithm with the existent network but they not ensure the optimality.

We used two data sets (one original and one artificial) in our paper with three algorithmic variants (exhaustive, greedy add, and greedy drop). For original data, all three algorithms have produced the same result. Therefore, we only show one optimum ranking in Table 2. However, these algorithms have produced different results (Table 4) for the artificial data set. We stated in the abstract that only exhaustive optimization will give the true optimum.

**Reply to comments of reviewer #4**

(reviewer comments in black, replies in blue)

Dear Authors, I've read your manuscript with enthusiasm, as this topic is of my great interest. I do agree with the authors that there is no consensus in the design of monitoring networks, but I do not agree with the conclusions found in this paper. In general, I find that the scoping of the problem validate the obtained results, and therefore, beats its purpose. This is further detailed in the comments below. In addition, I believe that critical methods and literature are overlooked, especially in the use of metaheuristics for the design of sensor networks. Also, I believe that the document could be better structured, as sometimes results and methodology sections overlap.

We appreciate your thoughtful comments and the amount of time spent to provide such constructive suggestions and discussions. In the following section, reviewer comments are in black color, author responses are in red color, and changes planned to be made to the text in the manuscript are in *italics* and underline font.

We agree with you on many aspects of monitoring network design, though our different opinions on our theoretical arguments have led us to different conclusions. In our opinion, the results from (we now clarified that) a case study can only serve as illustration and not for normative arguments for the use of a particular objective function. In information-theoretical network design, performance metric usually is optimization by one of the objective functions. Therefore, we believe choice of the objective functions must be justified by theoretical argument otherwise, evaluation would become circular. We believe you agree with that, when you say the scoping of the problem validates the results.

We therefore emphasized in several places in the paper that it is not the case study results that lead us to our conclusions (they are merely for illustration), but it is the arguments for scoping the problem and interpretation of the information measures that lead to our conclusions.

We limited the scope of our paper to a substantial body of literature employing several information measures as objective functions. While we agree that several different objective functions may be justified to reflect the users and their decision problems in a specific case study, we believe these should then be explicitly derived from those decision problems. In the case of government-funded general purpose monitoring networks, these explicit formulations are often not possible. It then makes sense to maximize the collected information by the network, without judgement on what that information is used for.

This maximizing the total collected information is apparently also the underlying objective of the several papers dealing with information-based monitoring network design that we discuss here. We assume this, because in those papers, there is typically no mention of a decision problem to motivate the objective functions.

The main point of our paper is that, while the idea of minimizing redundant information for monitoring networks intuitively makes sense to promote efficiency, we believe the only reason to justify this is in fact the underlying desire to maximize the non-redundant information the network collects. Redundancy in itself does not hurt, it only hurts through the information loss it causes, by decreasing the amount of non-redundant information collected.

This amount of non-redundant information is precisely what is measured by the joint entropy. In other words, the only reason to include some objective to minimize redundancy, is already covered within the single objective of maximizing joint entropy. .

While we do not claim that joint-entropy is the only objective needed in all situations, we do claim it is the only measure needed in absence of motivations that go beyond the idea of collecting maximum information.

We found many previous studies did not contain these explicit other motivations, and often language that suggests that the objective to minimize redundancy is motivated from preventing inefficient information collection, or is presented as self-evident.

Some examples of this language are given in the replies to specific comments.

On the greedy search strategies and metaheuristics: We found that most of the previous studies (listed in Table 1) used greedy search to find optimal ranking. We compared greedy and exhaustive approaches to raise awareness in the scientific community that greedy optimization might fall into local optimum; and to show why exhaustive optimization is not feasible in higher dimensions. We also clarify why no greedy approach could exist that is guaranteed to find all optimum sensor subsets. We now added that more explicitly in the conclusions.

We agree with you about merits in using metaheuristics as a search tool, and that it should be mentioned as an alternative. In this paper, we intend to keep focus on our main message (importance of choice of objective function). Indeed, comparing greedy, exhaustive and metaheuristic would be an interesting research question, and we now promote this idea in further work section, as we agree it was an oversight not to discuss these. Also, we improved the paper structure in several other places by following your suggestions and other reviewers'.

We added in section 4.3

*Another limitation of this comparison is that we did not consider metaheuristic search approaches (Deb, 2002; Kollat, 2008), which fall in between greedy and exhaustive approaches in terms of computational complexity, could serve to further explore the optimality versus computational complexity trade-off.*

We added in the conclusions:

*It remains to be demonstrated in further research how serious this loss of optimality is in a range of practical situations, and how results compare to intermediate computational complexity approaches such as metaheuristic algorithms.*

We added in further work:

*In this paper, we focused on the theoretical arguments for justifying the use of various objective functions, and compared a maximization of joint entropy to other methods, while using the same data set and quantization scheme. Since the majority of previous research used greedy search tools to find optimal network configurations, we compared greedy and exhaustive search approaches to raise awareness in the scientific community that greedy optimization might fall into local optimum, though its application can be justified considering computation cost of exhaustive approach. Banik et al. (2017) compared computation cost for greedy and metaheuristic optimization (Non-dominated Sorting Genetic Algorithm II). They reported that the greedy approach resulted in drastic reduction of the computational time for the same set of objective functions (metaheuristic computation cost was higher 58 times in one trial and 476 times in another). We recommend further investigation of these three search tools in terms of both optimality (for the maxJE objective) and computation cost.*

Objective functions for information-theoretical monitoring network design: what is optimal?
 General comments
What is the definition of optimality?
The rhetorical question "what is optimal?" in the title can be understood in two ways. Firstly it may refer to the question "what is the optimum network configuration?" (given that we have defined what we want from the network, i.e. the objective function). Secondly it might ask the question "what do we want from the network?"; "what do we consider optimal?"
Putting this question in the title was our way of drawing the reader's attention to the core difference between these two. Also it serves to hint at the fact that choice of objective function controls optimal answer—changing the objective function would change the subsequent optimal answer, so the objective function must be justified by reasoning independent of the numerical results. We will add a brief explanation in the beginning of the introduction.

Added in L23:
*In this paper, we do not answer the question ``what is optimal?'' with an optimal network. Rather, we reflect on the question of how to define optimality in a way that is logically consistent and useful within the monitoring network optimization context, thereby questioning the widespread use of minimum dependence between stations as part of the objectives.*

In the introduction you point out that monitoring objectives are posed as part of a wider decision problem (l18-19). Later, you state that the objective in the design of a sensor network is to maximise its joint entropy (l26-27) as sensor networks may support many decisions. At this point, this becomes a normative approach to the design problem, where you define what optimallity is, and that other objectives should be secondary (l347-348). As a consequence, it is clear that the problem is not a multi-objective optimisation problem anymore, and any tradeoff with other objectives (such as minimising redundancy) will directly reflect in a performance loss. This is actually seen in Table 2, where you clearly point it out.

In this paper we are not trying to argue that maximizing joint entropy is the final answer to optimization of monitoring networks. What we do argue is that choosing an objective function for an optimization that is intended to help decision makers, rather than describing their existing choices, necessarily has some normative elements to it, and should therefore be justified by arguments independent of numerical results.
We bring forward such theoretical arguments to argue that if maximizing the information collected by the network is the objective, then joint entropy is the mathematical expression to maximize. Minimizing redundancy only serves as one of the means to achieve that goal. When the goal itself is the objective function it is not necessary to add one of the means as a secondary objective. This will only lead to a loss in performance on the main goal (in a way by overly accounting for the effect of dependency).
Our scope in this paper is the information theoretical approaches to monitoring network design, which do not bring in external motivations to consider other objective functions, and as such we can only assume maximizing some idea of information is the objective. No other motivations for minimum redundancy were found in the previous literature.
So you are correct that at least part of our approach is normative, in the sense that we argue against use of minimum dependency as an objective. We think that was also clear from our wording, but made some small textual modification to make it clearer.

Defining the design problem in these terms makes it sufficiently narrow to justify the use of single-objective optimisation, but the point is that not every problem is.
We agree that not every design problem is single objective. In absence of motivations based on decision problems supported by the network, "optimization of collected information" could be a reasonable objective. Our main goal of this paper is to clear up the misunderstanding that minimum redundancy needs to be explicitly added as a separate objective to achieve an efficient network. We argue that the rationale for including minimum redundancy is missing.

This also connects with the three main arguments presented in the motivation (s 1.2). l44. "Firstly, we argue that objective functions for optimizing monitoring networks can, in principle, not be justified by case studies" - I do not agree with this postulate. The objective of monitoring is to provide information about the state of a system, to support a given action. Measuring for the sake of measuring do not serve any purpose.

l48. "However, from case studies, we cannot draw any normative conclusions as to what objective function should be preferred." - Preferences are not normative, but relative to the decision problem, objectives and context. These are particular to each case study.

We agree that the objective functions should be informed by the decision problem, objectives and context. We realize now that, to accurately convey what we mean, the sentence should be: "However, from **the results of** case studies, we cannot draw any normative conclusions as to what objective function should be preferred". From your comments it looks like you agree this would be circular.

We do agree on the benefit of case studies to formulate case-specific objectives that are motivated by the stakeholders in the monitoring network. Eliciting these objectives would be a valuable use of case studies. However, that is not what the case studies are used for in the papers we discuss here. We also make our point about case studies to make it explicit to our readers that the case study in our paper should not be interpreted as a (circular) argument for use of joint-entropy, but merely as an illustration of the different objectives at work. The arguments for using joint entropy are given before and after the case study, but do not rely on its results.

Measuring for the sake of measuring indeed makes no sense, but if we are measuring for an unknown "set of sakes", then maximizing information content gives maximum potential for use. Though it is possible to get information from the network without utility for decision, it is impossible to have utility from the network for decisions without it providing information.

We edited the first argument as follows:
*Firstly, we argue that objective functions for optimizing monitoring networks can, in principle, not be justified by analysing the resulting networks from application case studies. Evaluating performance of a chosen monitoring network would require a performance indicator which in itself is an objective function. Case studies could be helpful in assessing whether one objective function (the optimization objective) could be used as an approximation of another, underlying, objective function (the performance indicator). However, from results of case studies we should not attempt to draw any conclusions as to what objective function should be preferred. In other words: the objective function is intended to assess the quality of the monitoring network, as opposed to a practice where the resulting monitoring networks are used to assess the quality of the objective function.*

l50. "Secondly, we argue that the joint entropy of all signals together is in principle sufficient to characterize information content and can therefore serve as single optimization objective" - I do agree with this postulate, as long as the objective is to maximise joint entropy. This goes back to the first comment.

Yes, our argument is that maximizing joint entropy is the only objective needed to maximize joint information content, if no other requirements are given about a target for prediction or a decision problem. Would you also agree with the statement if we said: "joint entropy characterizes joint information content without the need to separately account for redundancy."? This is what we argue and disagree with much of the literature. We now worded argument 2 as follows:
*Secondly, we argue that, in purely information-based approaches, the joint entropy of all signals together is in principle sufficient to characterize information content and can therefore serve as single optimization objective. Notions of minimizing dependence between monitored signals through incorporation of other information metrics in the objective function lack justification and are therefore not desirable.*

l53. "Thirdly, multi-objective approaches that use some quantification of dependency or redundancy as a secondary objective, next to joint entropy, could only be justified if redundancy is interpreted as beneficial for creating a robust network" - I do partially agree with this postulate, but then again is linked to the definition of optimality. Given the decision

problem, a decision-maker may opt to trade some improvement in joint entropy for redundancy (as an example), and that is out of the scope of what this paper presents.

We agree, that the pareto front defined by a max JE, max Redundancy multi-objective problem could be interesting to explore. Our main point here was, however, that there is no justification for adding **Minimum Redundancy** as an objective. To emphasize this, we actually suggest that the opposite, **maximizing redundancy**, makes sense in certain contexts. Perhaps this was not the most clear way to formulate this and we now rewrote the third point as following:

*Thirdly, because the undesirable information inefficiencies associated with high dependency or redundancy are already accounted for in maximizing joint entropy, we could actually argue for maximizing redundancy as a secondary objective, because of its associated benefits for creating a network robust against failures of individual sensors. Minimization of redundancy would mean that each sensor becomes more essential, and therefore the network as a whole more vulnerable to failures in delivering information. Adding a trade-off with maximum redundancy is outside the scope of this paper, but serves to further illustrate the argument against use of minimum redundancy.*

 In general, once you assume that joint entropy corresponds to the definition of optimality, the problem is self-validated. Leading to the conclusions that you are presenting such as: "Information theory provides a valid framework for monitoring network design" (l344), and that single-objective optimisation is sufficient to approach the sensor network design problem.
We do agree that the sentence you quoted is not directly a conclusion from the main arguments presented is this paper, and hence we removed it. To cl
We do not assume that JE corresponds to the definition of optimality, but we argue that, within the scope of the 20+ papers that use minimization of dependency related information measures, JE is sufficient, since no motivation for reducing redundant information was given, except implicitly as a means to capture more information (see quotes below)..
We are aware of the self-validation problem, and this is why we emphasized at several points that our case study should not be interpreted as validation of our points, but the validation should be sought in our demonstration of the meaning of information measures and how they relate to each other.

We will make it clearer at the beginning of the conclusions that these come from  interpretation of the measures, not from the case study results..

Added in L387:
*The aim of this paper was to contribute to better understanding the problem of optimal monitoring network layout using information-theoretical methods. Since using resulting networks and performance metrics from case studies to demonstrate that one objective should be preferred over the other would be circular, the results from our case study served as an illustration of the effects, but not as arguments supporting the conclusions we draw about objective functions. We investigated the rationale for using various multiple-objective and single-objective approaches, and discussed the advantages and limitations of using exhaustive vs. greedy search.*

The justifications for including minimum redundancy-related measures that we found in previous literature, apart from the quote in our paper from Mishra and Coulibaly [2009], are in the following quotes:

- Alfonso et al 2010b: *"The main contribution of this paper is that joint entropy and total correlation are independent objectives that must be optimized.".*  --- Note that we show they are not independent. They argue redundancy should be minimized to find an independent set of stations. They stated this argument comes from [Mishra and Coulibaly,2009], and they proposed to use total correlation instead of transinformation to achieve that goal.

- Li et al (2012): "*highest information content and **avoid dependent stations as much as possible**, guaranteeing while the stations within and outside of the optimal set has high common information*". They also argued "*The information-redundancy tradeoff weights **provide the user a flexible handle to balance the two conflicting objectives**: maximum information and minimum redundancy.*"

- Keum and Coulibaly (2017) rephrase goal of independent network: "*Therefore, the amount of duplicated or sharable information in a network explains the redundancy or **ineffectiveness of the network**.*". They also comment on MIMR: "*On the other hand, the MIMR reformulate the multiobjective problem to a single objective optimization by merging the different criteria into one objective using weighting factors [e.g., Li et al., 2012; Fahle et al., 2015]. However, the weight for each objective should be assumed in advance. In this study**, the former approach is applied not to make any prior assumptions** but to compare various optimal networks in decision processes.*".

- Banik et al. (2017) stated TC as single objective should not be used: "*Minimizing this objective means reducing the correlated information. The objective of the problem being to maximize the information furnished by the sensors, the **TC function is considered always in combination with JH**. In fact, TC as a single objective furnishes solutions with less-correlated sensors, for example, terminal nodes, **with a poor content of information**.*"

Note that all these argue that stations need to be independent, but do not give a reason why, except for effectiveness, which we argue is covered by looking at the total non-redundant information the  network delivers.

Below are some of the 20+ papers that we referred to earlier in this reply:

(1)      Wang, W.; Wang, D.; Singh, V. P.; Wang, Y.; Wu, J.; Zhang, J.; Liu, J.; Zou, Y.; He, R. Information Theory-Based Multi-Objective Design of Rainfall Network for Streamflow Simulation. *Advances in Water Resources* **2020**, *135*, 103476. https://doi.org/10.1016/j.advwatres.2019.103476.

(2)      Li, H.; Wang, D.; Singh, V. P.; Wang, Y.; Wu, J.; Wu, J.; He, R.; Zou, Y.; Liu, J.; Zhang, J. Developing a Dual Entropy-Transinformation Criterion for Hydrometric Network Optimization Based on Information Theory and Copulas. *Environmental Research* **2020**, *180*, 108813. https://doi.org/10.1016/j.envres.2019.108813.

(3)      Werstuck, C.; Coulibaly, P. Assessing Spatial Scale Effects on Hydrometric Network Design Using Entropy and Multi-Objective Methods. *JAWRA Journal of the American Water Resources Association* **2018**, *54* (1), 275–286. https://doi.org/10.1111/1752-1688.12611.

(4)      Wang, W.; Wang, D.; Singh, V. P.; Wang, Y.; Wu, J.; Wang, L.; Zou, X.; Liu, J.; Zou, Y.; He, R. Optimization of Rainfall Networks Using Information Entropy and Temporal Variability Analysis. *Journal of Hydrology* **2018**, *559*, 136–155. https://doi.org/10.1016/j.jhydrol.2018.02.010.

(5)      Banik, B. K.; Alfonso, L.; Di Cristo, C.; Mynett, A. Evaluation of Different Formulations to Optimally Locate Sensors in Sewer Systems. *Journal of Water Resources Planning and Management* **2017**, *143* (7), 04017026. https://doi.org/10.1061/(ASCE)WR.1943-5452.0000778.

(6)      Alfonso, L.; Lobbrecht, A.; Price, R. Optimization of Water Level Monitoring Network in Polder Systems Using Information Theory. *Water Resources Research* **2010**, *46* (12). https://doi.org/10.1029/2009WR008953.

(7)      Li, C.; Singh, V. P.; Mishra, A. K. Entropy Theory-Based Criterion for Hydrometric Network Evaluation and Design: Maximum Information Minimum Redundancy. *Water Resources Research* **2012**, *48* (5). https://doi.org/10.1029/2011WR011251.

(8)      Fahle, M.; Hohenbrink, T. L.; Dietrich, O.; Lischeid, G. Temporal Variability of the Optimal Monitoring Setup Assessed Using Information Theory. *Water Resources Research* **2015**, *51* (9), 7723–7743. https://doi.org/10.1002/2015WR017137.

(9)      Samuel, J.; Coulibaly, P.; Kollat, J. CRDEMO: Combined Regionalization and Dual Entropy-Multiobjective Optimization for Hydrometric Network Design. *Water Resources Research* **2013**, *49* (12), 8070–8089. https://doi.org/10.1002/2013WR014058.

(10)     Stosic, T.; Stosic, B.; Singh, V. P. Optimizing Streamflow Monitoring Networks Using Joint Permutation Entropy. *Journal of Hydrology* **2017**, *552*, 306–312. https://doi.org/10.1016/j.jhydrol.2017.07.003.

(11)     Keum, J.; Coulibaly, P. Information Theory-Based Decision Support System for Integrated Design of Multivariable Hydrometric Networks. *Water Resources Research* **2017**, *53* (7), 6239–6259. https://doi.org/10.1002/2016WR019981.

(12)     Keum, J.; Coulibaly, P.; Razavi, T.; Tapsoba, D.; Gobena, A.; Weber, F.; Pietroniro, A. Application of SNODAS and Hydrologic Models to Enhance Entropy-Based Snow Monitoring Network Design. *Journal of Hydrology* **2018**, *561*, 688–701. https://doi.org/10.1016/j.jhydrol.2018.04.037.

(13)     Huang, Y.; Zhao, H.; Jiang, Y.; Lu, X. A Method for the Optimized Design of a Rain Gauge Network Combined with Satellite Remote Sensing Data. *Remote Sensing* **2020**, *12* (1), 194. https://doi.org/10.3390/rs12010194.

(14)    Banik, B. K.; Alfonso, L.; Torres, A. S.; Mynett, A.; Di Cristo, C.; Leopardi, A. Optimal Placement of Water Quality Monitoring Stations in Sewer Systems: An Information Theory Approach. *Procedia Engineering* **2015**, *119*, 1308–1317. https://doi.org/10.1016/j.proeng.2015.08.956.

(15)    Banik, B. K.; Alfonso, L.; Di Cristo, C.; Leopardi, A. Greedy Algorithms for Sensor Location in Sewer Systems. *Water* **2017**, *9* (11), 856. https://doi.org/10.3390/w9110856.

(16)    Keum Jongho; Coulibaly Paulin. Sensitivity of Entropy Method to Time Series Length in Hydrometric Network Design. *Journal of Hydrologic Engineering* **2017**, *22* (7), 04017009. https://doi.org/10.1061/(ASCE)HE.1943-5584.0001508.

(17)    Leach, J. M.; Coulibaly, P.; Guo, Y. Entropy Based Groundwater Monitoring Network Design Considering Spatial Distribution of Annual Recharge. *Advances in Water Resources* **2016**, *96*, 108–119. https://doi.org/10.1016/j.advwatres.2016.07.006.

(18)    Leach, J. M.; Kornelsen, K. C.; Samuel, J.; Coulibaly, P. Hydrometric Network Design Using Streamflow Signatures and Indicators of Hydrologic Alteration. *Journal of Hydrology* **2015**, *529*, 1350–1359. https://doi.org/10.1016/j.jhydrol.2015.08.048.

(19)    Maymandi, N.; Kerachian, R.; Nikoo, M. R. Optimal Spatio-Temporal Design of Water Quality Monitoring Networks for Reservoirs: Application of the Concept of Value of Information. *Journal of Hydrology* **2018**, *558*, 328–340. https://doi.org/10.1016/j.jhydrol.2018.01.011.

(20)    Pádua, L. H. R. de; Nascimento, N. de O.; Silva, F. E. O. e; Alfonso, L.; Pádua, L. H. R. de; Nascimento, N. de O.; Silva, F. E. O. e; Alfonso, L. Analysis of the Fluviometric Network of Rio Das Velhas Using Entropy. *RBRH* **2019**, *24*. https://doi.org/10.1590/2318-0331.241920180188.

(21)    Vivekanandan, N. Evaluation of Stream Flow Network Using Entropy Measures of Normal and Lognormal Distributions. *Bonfring International Journal of Industrial Engineering and Management Science* **2012**, *2* (issue 3), 33–37. https://doi.org/10.9756/BIJIEMS.10040.

(22)    Vivekanandan, N.; Jagtap, R. S. Evaluation and Selection of Rain Gauge Network Using Entropy. *J. Inst. Eng. India Ser. A* **2012**, *93* (4), 223–232. https://doi.org/10.1007/s40030-013-0032-0.

(23)    Wang, W.; Wang, D.; Singh, V. P.; Wang, Y. Spatial-Temporal Evaluation of Rain-Fauge Network Based on Entropy Theory. In *EPiC Series in Engineering*; EasyChair, 2018; Vol. 3, pp 2293–2300. https://doi.org/10.29007/1kc9.

(24)    Xu, P.; Wang, D.; Singh, V. P.; Wang, Y.; Wu, J.; Wang, L.; Zou, X.; Chen, Y.; Chen, X.; Liu, J.; Zou, Y.; He, R. A Two-Phase Copula Entropy-Based Multiobjective Optimization Approach to Hydrometeorological Gauge Network Design. *Journal of Hydrology* **2017**, *555*, 228–241. https://doi.org/10.1016/j.jhydrol.2017.09.046.

(25)    Werstuck, C.; Coulibaly, P. Hydrometric Network Design Using Dual Entropy Multi-Objective Optimization in the Ottawa River Basin. *Hydrology Research* **2017**, *48* (6), 1639–1651. https://doi.org/10.2166/nh.2016.344.

What are the alternatives for solving the optimisation problem?

From the methodological point of view, I see that you propose the greedy drop algorithm, as an alternative to greedy selection, especially attractive in the cases where exhaustive search is not feasible (l216-223). However, you omitted mentioning metaheuristics to approach this

issue. These are often used in the design of monitoring networks when the combinatorial problems are too large. This being the argument for greedy approaches.

We fully agree with the reviewer on this one. We now mention metaheuristic approaches, and put in a few words about the complexity of the search space. Metaheuristic approaches indeed fall somewhere in between greedy approaches and exhaustive search. It would be interesting to explore the tradeoff between computational complexity and optimality. We mention this where greedy approaches are discussed and in the future work section. See the quotes of added text earlier in this reply.

What are the cases of sensor network design that are considered?

When designing a monitoring network, you may have one of 3 possible scenarios: augmentation, reduction or relocation. Augmentation makes for the case when additional sensors are to be placed in the network. Reduction, accounts for the case when sensors would like to be removed from the network. Relocation, deals with the issue of changing the placement of the sensors. The case you illustrate in this paper corresponds to reduction, as the objective is to select a sub-set of sensors from a larger pool. However, you use justifications from the augmentation case (l221-222) to justify the use of a greedy algorithm. I think it is necessary to better define the optimisation problem in this respect.

Yes, we agree on those 3 scenarios. Note that, in this paper we show that to maintain optimality, reduction or augmentation may both need to go hand in hand with relocation. This is why no greedy approach to step by step expansion or reduction of sensors can exist that can be guaranteed to be optimal. We now added the case of reduction as well to line 221-222, since it also applies to that scenario.

About the definition of the optimization problem: If we keep the focus on information theoretical objective functions, which is the scope of this paper, then we will need data for the time series at sensor locations to calculate the information measures, whether they are candidates for removal or potential new sites.
The choice of interest in either augmentation, reduction or relocation case mainly influences the way we get the data (from measured time series vs. modeled ones), and the constraints on which locations we consider as candidates. Even in case of network expansion/augmentation,, stations will need to be selected from a pool modeled locations. In rain gauge network design, for example, gridded radar or satellite data can be assumed as hypothetical sensors to solve augmentation or relocation of existing network. For example, Yeh et al (2017) used simple entropy-based objective function to solve network augmentation. In terms of requirements for the objective function, there is no difference between these problems.

We do not argue that this is the only way to formulate the optimization problem, but given the amount of discussion the choice of objective function in the information-theoretical setting already raises, we keep our focus on that aspect. We made several smaller edits throughout the text to make the points above clearer.

Yeh, H.-C.; Chen, Y.-C.; Chang, C.-H.; Ho, C.-H.; Wei, C. Rainfall Network Optimization Using Radar and Entropy. *Entropy* **2017**, *19*, 553.DOI: **https://doi.org/10.3390/e19100553**

Specific comments

 The section on presenting the basics of information theory can be better summarised. There is plenty of "well-known" material on it.

Since not all of the HESS audience is necessarily aware of information theory, and the interpretation of information measures is at the core of the debate here, we believe the inclusion of some well-known material is justified. For example, reviewer #1 states: "I enjoyed reading the manuscript, especially the introduction to the information theory terms.".

Though readers like you, who have a deep background and opinion on the information measures is definitely are a  key part of the intended audience we hope to convince, we also hope to serve a wider audience who would welcome some introductory material.

We deleted a few points that are not directly relevant to the argument here, such as the different units, and the thermodynamic origin of the entropy concept.

 l172-173 requires a reference.

Thanks for catching this, it is fixed by adding the following references.

added in L171:

Maximizing network information content, through either the sum of marginal entropy or joint entropy, is the common theme among existing methods *(Alfonso et al., 2010b; Li et al., 2012; Samuel et al., 2013; Keum and Coulibaly, 2017; Wang et al., 2018; Huang et al., 2020).*

l187 requires explanation about what is objective GR3

Thanks for catching this, we did explain GR1 to GR6 in the appendix but forgot to direct our readers there. We also noticed a critical word "not" that we forgot to insert. it is now modified:

Added in L187:

*This is equivalent to the GR3 objective proposed by Banik et al. (2017), as part of six other objectives (see appendix B for more detail) proposed in the same paper, which did not provide preference for its use.*

l200-201 These are not MOO methods. These are objective functions.

We agree. We replaced  "three multi-objective optimization methods" with "three other (sets of) objective functions from previously proposed methods".

l202- 203 requires references

Thanks, it is properly referenced now.

l218-220 It is not true that the only way of selecting stations is using greedy algorithms.

We agree with you that the greedy algorithm is not the only way. But, a greedy approach is used by the majority of studies in the literature. We modified the text to clarify this.

We changed as follows:

*In the majority of existing literature listed in Table 1, one constraint has often implicitly been imposed: to treat the selection of stations as greedy optimization, meaning that one station is added to the set of selected stations each time while trying to optimize the objective function, without reconsidering the already selected stations in the set.*

l221 is not combinatorial "explosion". Instead we can argue that the problem is exponentially complex (Oˆn)
Will be modified as recommended.

We agree that the the search space for evaluating all sub-networks (the power set of the set of sensor locations) grows exponentially with 2^n. The greedy approach has a search space of O(n^2).

We added::

*A practical reason for this is numerical efficiency; an exhaustive search of all subsets of k stations out of n possible stations will need to consider a large number of combinations, since the search space grows exponentially with the size n of the full set of sensors (2^n combinations of sensors need to be considered).*

l222 I think here you are mixing two design problems. One of the problems is of design (where to measure at several locations), and other of augmentation (Where to put additional stations). Of course these are clearly different processes.

You are correct that technically these are different problems where perhaps in one case, you would install a new sensor, and in the other, you would just switch it on. In this research, we are just focusing on the benefits side (the information obtained), and how to quantify that. While the pool of candidate locations and way to model what could be measured or is measured may be different, the way to formulate the objective functions would be the same. So the problems are indeed different, but not that much in the aspects that we consider in this paper. See also the answer for "What are the cases of sensor network design that are considered?"

l232 I think it should be necessary to include methods using metaheuristics for comparison.

As mentioned in our answer to comment 1, we will discuss the metaheuristic approach in the further work section.

We agree with the reviewer that for a full investigation of computationally efficient methods, we would need to include the metaheuristic approaches. We mention this in the future work section. For our current paper, we just want to highlight one other very computationally cheap greedy approach - O(n^2) - , which we prove non-optimal by counter example. We then use that to briefly discuss why greedy approaches cannot be optimal, but also what the implication for expanding or reducing a network is.

It would indeed be interesting to look at approaches of intermediate computational complexity and see how much optimality can be gained compared to either of the the greedy approaches.

l233 It is not clear what logistical reasons are. Should not these be included in the optimisation constraints?

We agree this was vague. We now clarified by changing as follows:

*In this comparison, we will investigate whether the exhaustive optimization yields a series of networks where an increase in network size may also involve relocating stations. This may not always be practically feasible or desired in actual placement strategies, where networks are slowly expanded one station at a time. Occurrence of relocation in the sequence of growing subsets would also show that no greedy algorithm could exist that guarantees optimality.*

l235-236 This seems speculative at this point, and better be moved to other place in the document (perhaps introduction?)

Agreed. This is a hypothesis that we falsify in the paper, and we removed it here .

l233 This "golden standard" expression seem somewhat loose talk. Can just point out is the only way to prove optimality?

Agreed. "golden standard" expression is replaced by "optimality benchmark"

Figure 4 can be improved. labels are hard to read, and would be more informative just to keep the ID's?. Also if its for monthly data, have you considered using flow duration curves instead? alternatively, please consider using a log y-scale, as discharge distributions are positively skewed. In addition, y-scale label is missing.

Thanks for the idea, we  modified fig 4.

We changed y-scale to the log scale. The box plot and statistics show much of the information that can be presented by FDC. We tried the FDC, but it becomes quite busy. Also, the information measures are more directly linked to the pdf than to the cdf, so we find the stick to the current figure with the logarithmic y axis you suggested.

in Table 2 are presented the results of different optimisation criteria (defined as MOO), but there is no indication of the selecting strategies, thus the information of the whole Pareto set is unavailable. In top, that is the whole point of MOO, that there exists trade-offs between objectives that cannot be assessed by the modeller.

We agree with the reviewer about the role of trade-offs weights in MOO. But, we chose to accept Li et al. (2012)'s conclusion on insignificant effect of information-redundancy trade-off weights in this particular case study. They conducted sensitivity analysis and reported that MIMR results are stable with respect to trade-offs weights in MOO. We compared maxJE with two other highly cited single-objective methods in our field because they either directly minimize redundancy (minT) or indirectly minimize redundancy by imposing constraint in search space (WMP). We didn't discuss trade-offs weights and presented Li et al. (2012)'s conclusion since we argue that this trade-off is irrelevant in the first place. As you correctly pointed out, the subjectively issue can be raised on the modeler's decision of trade-off weights. Therefore, we decided to present the Li et al. (2012) results in our paper for the sake of having unbiased comparison, assuming they represent the use of the method as intended.
We added:
*For the multi-objective approaches used in the case study, we used the same weights as the original authors to identify a single solution.*

S4.2 includes parts that should have been presented in the methodology and not in the results section.

We will see if we can re-organize some material to improve the flow. This is somewhat challenging because the case study serves as illustration, not as an experiment needed to support conclusions. To explain our rationale for including this material here: This section is meant to further explore our key point that minimizing dependence is not needed as an objective. This is a result from the reasoning in the paper, which is continued in the results section so it can benefit from numerical results for illustration. We bring in the pareto space plot to illustrate that the maximizing redundancy objective changes the pareto front considered. We see that as an illustration of the discussion, rather than a predefined experiment to test a hypothesis, hence it's placement here. We looked at this carefully and moved some material.

l276. H(X,X) = H(X), therefore two completely "dependent" are exactly as informative as one.

Your statement on two completely "dependent" is correct and sharper than our original formulation. We modified it as suggested.

Visualisation using Venn diagrams are an excellent way of presenting concepts, but are really hard to follow to describe precise quantities. Will it be possible to re-think Figure 5 in a simpler manner?
We agree on the loss of precise quantities, but we think the figure as it is provides a good visual overview of how the info measures vary, by growing and shrinking colors. Many other ways of plotting lose the interconnections between the measures plotted. To have the best of both worlds, we pointed to the code repository in which all data behind that figure and tables can be found, so the precise quantities are available for further study.

l287-288 Those are precisely the trade-offs that decision-makers do in MOO, and the reason of its relevance. If you claim that maximum joint entropy is somewhat equivalent to minimum

total correlation, then these two objective functions are not conflicting, and therefore, by maximising one, you are maximising the other. Precisely as shown in Figure 7.

In these lines we indicate that we do see the relevance of MOO in the case of **maximizing** total correlation next to maximizing joint entropy. However, the papers cited in this work **minimize** total correlation as a secondary objective. As seen from figure 7, this will reduce the joint entropy and, as argued in the text, it is doing so without independent justification for that secondary objective.

Low total correlation is one of the factors in max joint entropy, but they are not equivalent. There is indeed some correlation between the objectives, but there is still some trade off (as indicated by the dashed pareto front in figure 7, which previous approaches explored). In this paper, we argue that this trade-off is not relevant, and that an exploration of another pareto front in this space, could be justified from the point of view of maximizing robustness against failure.

l297-298 The whole point in obtaining the Pareto front between maximum joint entropy and minimum total correlation explores the trade-offs between a network that is able to capture most information, vs a network that has little "information overlap", which is not the same as the individual entropies are different. Therefore, these are different metrics, and is the reasoning behind finding the Pareto set in the dashed line of Fig 7.

We agree that this is a good summary of the justification given in most previous literature, and we do agree there is a trade-off, **but what is the motivation behind having little information overlap if it's not to capture more information** (which is already addressed by joint entropy)? We did not find such motivation or justification in the previous literature and don't believe there is one.

Hence this is the core point of our paper, which results from reasoning about the information measures and not from case study results. The case study results just illustrate the effects. We did subtle rewording throughout the paper to clarify this point.

S4.3 A lot of the text in this section can be part of the methodology.

We agree that some material can be moved to methodology. We moved some material around to best serve the clarity of our messages, keeping in mind that we think the points here are secondary in importance to our points about the objective functions.

Table 4 is quite hard to read. Also, this table is precisely showing you that optimallity is not found using greedy algorithms.

Indeed, the point of including table 4 and 5, for which we generated a dataset, is to show that no greedy algorithm can exist that is guaranteed to find the optimum. We now included this in the caption to make that clearer, and also reorganized table 4 (now 5) slightly to make it easier to read. We split rows of exhaustive optimization into added and removed stations), and improved layout, to make the table easier to read..

Table 5 Be consistent in the amount of significant digits through your document

Thanks for catching this. We will make sure to be consistent at least within each table. We felt the need to include extra digits in this table, to show the small differences that occur, and

hence provide a falsification of the hypothesis that greedy algorithms always lead to optimal solutions. Even a small difference would prove that.

l344 No information system is justified if there is not objective to tackle. What problem am I addressing if I do not know what the problem is.

We reformulated to make it clearer what we mean: When the decision problem that is supported by the monitoring network is not simple and explicitly defined, and not fully known, then we may choose to maximize the information content of the network, as that provides potential value for the widest possible set of uses. You cannot derive value from a network that provides no information. On the other hand, giving information does not guarantee value for every decision.

An example is a hydrometric network of a national institute, which is maintained by government money to support a variety of a priori unknown decision problems.

We agree that attempts could be made to quantify an estimated value regardless, but our focus here is to discuss purely information based methods, for which there is a wide existing literature. We did significant rewording of the whole conclusions section.

l361 Large optimisation problems are tackled using metaheuristics, and has been a widely used approach. This has not been mentioned here at all.

Thank you for noticing this, we now mention metaheuristics as a potential other approach to address large problems. This is certainly worth further investigation in future work.

l364 the differences between the greedy and exhaustive search approaches have not been presented quantitatively. In this problem they may seem "little" (not being explicit about what little or big is in the context of the problem), but this cannot be ensured for larger problems that the ones presented here.

The differences between these approaches have been presented quantitatively in table 5. We agree this cannot be ensured for larger problems, hence our call for further research on this. We modified line 361 to clarify that we only speak about our simple case study and not in general when we say differences were small.

In our specific case studies, differences between exhaustive and greedy approaches were small; especially when using a combination of the greedy add and greedy drop strategy. It remains to be demonstrated in further research how serious this loss of optimality is in a range of practical situations, and how results compare to intermediate computational complexity approaches such as metaheuristic algorithms.

l367-368 Language has to be precise (how to numerically calculate this objective function, or other objective functions used in other approaches)

Agreed, we will modify as follows:

*Another important question that needs to be addressed in future research is to investigate how the choices and assumptions made (i.e., data quantization which influences probability distribution) in the numerical calculation of objective functions would affect network ranking.*

l370 Any information metric is hard to calculate with limited data.
Yes we agree, but the hardness (or data need) exponentially grows with the number of dimensions in the PDF. Entropy and pairwise mutual information are not a big problem, but total correlation and joint entropy become problematic for a large number of sensors. We now clarified as follows:

*These probability distributions are hard to reliably estimate from limited data, especially in higher dimensions, since data requirements grow exponentially.*

l378-379 I completely agree with this line "before thinking about how to optimize, we should be clear on what to optimize".
We are glad the reviewer agrees, this is also a reason why we placed the greedy vs non greedy discussion later in the paper as it is secondary to the main points about which objective function to use. We reorganised some material, without breaking the flow of the main argument.

l381 I visited the GitHub repository, but I was unable to find the code to reproduce these results. Only a reply to a WRR paper of 2018, and a fork of pysheds.
We apologize. They will be posted shortly.

[revised manuscript text omitted]

---

## Referee Report (RR1)

**Reply to review for Objective functions for information-theoretical monitoring network design: what is optimal? by Hossein Foroozand and Steven V. Weijs**

The Authors replied to the comments convincingly and precisely. Every comment was taken into account and properly explained. The changes were made improved the paper.

The final comment is that probably the title of the paper deceives a little bit: "what is optimal" leads to think to something more complete on the problem while as authors stated in the abstract, "only exhaustive optimization will give the true optimum". However, I recommend the article for the publication.

---

## Author Response (AR2)

**Reply to comments on revision 1 by reviewer #4**

(reviewer comments in black, replies in blue)

Dear authors, thank you for your reply.

I enjoyed seeing how this paper has turned into a better version of itself. I think your clarifications with respect to the points I initially highlighted were mostly met in your reply. To simplify the process, If I agree with your reply, I wont add it here. However, there are still some open points that I would like to further discuss based on the reply and re-structuring of the paper. I will start with the reply to the comments you made, then move to general comments about the paper (in its current state), and finally some specific comments line-by-line.

We thank Reviewer#4 for his/her positive review and constructive suggestions, which allowed us to improve the quality of the manuscript. In the following section, reviewer comments are in black color, author responses are in blue color, and changes proposed for the text in the manuscript are in *italics* and underline font.

\*\*\*\*\*\*\*\*\*\*\*\*\*\*\*\*\*\*\*\*\*\*\*\*

About the reply

\*\*\*\*\*\*\*\*\*\*\*\*\*\*\*\*\*\*\*\*\*\*\*\*
* * *
In the reply regarding the discussion about optimality you responded:

"In this paper, we do not answer the question ``what is optimal?'' with an optimal network.
Rather, we reflect on the question of how to define optimality in a way that is logically consistent and useful within the monitoring network optimization context, thereby questioning the widespread use of minimum dependence between stations as part of the objectives."

With this in mind, I think the title of the paper, beyond catchy, may be misleading. I am not against making this the point of discussion of your paper, but I believe that the title can better reflect the discussion that is inside. I think the main focus of discussion in this paper is to avoid using redundancy metrics unless the problem is explicitly defined to require them, and to see what is the effect of greedy-drop algorithm in the defining monitoring networks.

We made a subtle change to the title. By putting optimal in quotes, we refer to the word, i.e. the definition of optimal, rather than the concept. This makes clear that we discuss issues with the definition of optimality by the objective functions.
* * *
"l367-368 Language has to be precise (how to numerically calculate this objective function, or other objective functions used in other approaches)
Agreed, we will modify as follows: Another important question that needs to be addressed in future research is to investigate how the choices and assumptions made (i.e., data quantization which influences probability distribution) in the numerical calculation of objective functions would affect network ranking."

In this comment, I was also trying to highlight the fact that not all of the equations to calculate the metrics presented in your study were explicitly shown. At the moment, some of them are there, but I was not able

to find (in the body of the document) the complete information to replicate your study. In your paper your mention that you are using the same weights as the original authors (l259-260) to identify a single solution in the case of MOO, but it would be best if this is directly presented in the formulation to help replicate this study.

Thanks for pointing this out. We now add the following information about trade-off weight in MOO method to the caption of Table 2 where the results were presented. Also, we modified equation 8 to show how li et al. (2012) applied the trade-off weight to the MOO method.

Added in Table 2 caption:
*Note that  MIMR's trade-off weight ( $\lambda 1 = 0.8$ ) is based on the recommendation of  Li et al (2012) for this dataset.*
* * ** * *
About document in general
* * *
I think the general structure of the abstract can be improved. I suggest you being more concise on the methods and conclusions of your study. Also, I think it would be a better to end the abstract highlighting the conclusions rather than mentioning that there is a case study.
Thanks for your suggestion. We removed that sentence at the end of the abstract to better emphasize the paper's core message. We also sharpened the formulation about the impossibility of a greedy approach.

The structure of the document has to critically revised. I think a literature review section would not hurt, as you are including these topics in the methodology. Also, I noticed that you moved the position of the Methodology section, and now include topics which should not be presented there. The experimental setup (which experiments and using which data is being used to get which values) is not really clear; also, the first reference to the "synthetic dataset" is shown in the results section. In addition, the first part of your results reads as conclusions from the literature rather than from your own numerical experiments; in addition, during the results section there are suggestions of further work which should be included in its respective section. Please, reconsider the order of the sections in your document as it needs a big overhaul.

We systematically overhauled the paper's section2 structure based on your specific points. In revision #1, we applied some structural changes to accommodate other reviewers' suggestions. Your views seem to be more aligned with our original manuscript, so a few of those were partly reversed to accommodate your suggestions. It seems that this is the closest we can come to convergence on the best way to present this research, which is slightly more complicated due to the role of the case study, which is not the main source of our conclusions, but rather an illustration.

Language has to be revised. Paragraphs tend to be too long and sometimes drift over more than 1 idea at the time. I am asking you to be more concrete in each paragraph, as I agree that the overall length of the document and the amount of information presented is adequate. Also, it is important to revise the structure of some sentences to avoid using as many commas. Finally, make sure to stick to only one term for the same element through the text (i.e. station).

Thanks for catching this, we revised long paragraphs and sentences with many commas. Also, We made sure that the term "station" is used where we are referring to the 12 stations in this case study.

We uniformly used measures when referring to entropy, mutual information and similar measures, consistent with Cover and Thomas and Shannon sources.
* * *
Specific points
* * *
l82 - you can be more specific which trade-offs are being interpreted.
Thank you for your comments. We modified the text to improve its clarity.

Added in L94:
Section 4 introduces the results for the various methods, and then discusses the need for multiple objectives, the interpretation of trade-offs *between redundancy and total information*,...

l87 - You start the section with an addition connector ("In monitoring network design, also ..."). This leaves the reader without a context.
Agreed. As you recommended in previous comments, we now move the scope subsection to the introduction section to have a proper context for our readers. We also modified the first sentence of the section to be clearer:
Added in L67:
 "*The information-theoretical approach to monitoring network design is not the only option, and other objective functions have also been used for this problem*."

l89 - Kriging (with capital K)
Was fixed as recommended.

l91 - "such as for example" pick one
Thanks for catching this; we modified the text and provided more information about the referenced example.

l93 - "on that topic" This is redundant
Was fixed as recommended.
l94 - "on spatially distributed observed" In your case study you do not have spatially distributed observations (such as radar or remote sensor products), but rather discrete observations along a stream (even if they are spatially distributed).
Thanks for catching this, we changed it to " observed data on one single variable in multiple locations"
l95 - which -> that
In this case, we intended this as a non-defining clause (we believe all these objective functions should in principle define what we want from the network). We kept which, but now emphasized that we talk specifically about objective functions for monitoring network design:
*Keeping this limited scope allows us to discuss the interpretation of these objective functions for monitoring network design, which formalize what we actually want from a network.*

l95-98 - (Furthermore ...) In the methodology section this has to be way more concrete

Agreed,this paragraph has more introductory language and is moved to the introduction section. Section 2.6 provides concrete explanation for this purpose.

l99-101 - This can be slightly rephrased to avoid as many commas.

Was fixed as recommended. Down to one comma now.

l103 - The discussion is not demonstrated, but rather carried out.

Thanks for catching this, we reformulated the sentence to "Our discussion is numerically illustrated by a case study using data from Brazos River in Texas, as presented in Li et al., (2012), to allow for comparisons", as that most accurately describes the role of the case study in relation to the discussion.

l104 - "as we will argue" This is the methodology section. Please focus the text of what has been done in this particular paper (therefore written in present tense).

Agreed, we use present tense now.

l121-123 - Sounds more like a line for the introduction and not as part of the methodology

Agreed. The whole subsection is moved to the introduction section.

S2.1 should not be part of the methodology. This is part of the literature review.

Agreed. The whole subsection is moved to the introduction section.

l125 - I am not sure if Shannon in particular talks about uncertainty. I think his work is more closely related to compressibility of data. Similar, but yet different.

There is an agreement in the literature that  Shannon(1948) developed information theory (IT) based on entropy. In its section 6 (CHOICE, UNCERTAINTY AND ENTROPY), Shannon talks about the concept of uncertainty as defined by entropy. Shannon writes the following just before deriving his Entropy measure:
"Can we find a measure of how much "choice" is involved in the selection of the event or of how uncertain we are of the outcome?"

The beauty of Shannon's work is that it essentially unifies these seemingly different concepts of uncertainty and (un)compressibility. See also the recent WRR debate paper:

Weijs, S. V., & Ruddell, B. L. (2020). Debates: Does information theory provide a new paradigm for earth science? Sharper predictions using Occam's digital razor. *Water Resources Research*, 56, e2019WR026471. https://doi.org/10.1029/2019WR026471

l131 - "For monitoring networks, the information each sensor provides through its observations (outcomes) is therefore linked to the uncertainty of those outcomes before measurement. These are quantified through the probability distributions of the data." This line is quite hard to read, please consider it for simplification.

Thank you for your comments. We modified the text as follows to improve its clarity, while keeping precision:
*"For monitoring networks, we are interested in the information content of the observations from all stations. The information content is equal to the uncertainty about the observations before measuring. The uncertainty is quantified through probability distributions that describe the possible observations, based on the data."*

l135 - "the placement"

Thanks for catching this, it is fixed.

l137 - I think this is also a good place to mention joint entropy as you will be coming back to this later on the paper

l137 - "a random variable"

Entropy and joint entropy are represented by the H symbol applicable for both a single variable or multiple variables depending on the context. We now clarify that more by stating that explicitly when introducing joint entropy. "Joint entropy (Eq.2), as an extension of entropy beyond a single variable, measures the number of questions needed to determine the outcome of a multivariate system."

l140 - " Objective functions are often composed from these basic expressions. Details of each expression are presented below." I think this can be removed

We deleted the line as recommended.

l144 - You changed the name here to Discrete marginal entropy. For example (l138, entropy; l142, Shannon entropy; l144, Discrete marginal entropy). Please be consistent

We removed "Shannon" from line 142. Entropy is a general term, and discrete marginal entropy is a more specific term to explain how entropy values are calculated (i.e., continuous or discrete probability distribution). This specification is explained in the preceding sentence. We clarified by adding : "*single discrete variable*" to the line before the equation is introduced.

l149 you mention is only about units, while in l151 you mention that is used in answering questions. Which one is then the case? I think the answers depend on the quantisation rather than the base of the algorithm. Please clarify this.

Bits give the expected value of the number of equivalent binary question answers needed to determine in which bin the outcome falls, for any choice of quantization (layout of the bins). Switching base of the logarithm or unit, changes the type of question allowed to questions with e.g. more than two possible answers, and therefore the number of questions changes.There is also the connection to file sizes that can prove useful. We modified the text to clarify this point.

Added in L122:

*The choice of the logarithm's base for entropy calculation is determined by the desired unit — other information units are "nats" and "Hartley" for the natural and base 10 logarithms, respectively. For monitoring network design, the logarithm of base 2 is common in the literature since it can be interpreted as the needed number of answers to a series of binary questions, and allows comparisons with file sizes in bits; see e.g. Weijs et al. (2013a).*

l152-153 - This does not say much. Can you be more specific. Also, will you please generalise all of the equations in this section (H, H|X, T and C).

We provided a list of symbols in the appendix to provide a general overview. In the original draft, we generalized and explained all questions in one concise paragraph. But other reviewers requested to break it down and explain each equation right after the first appearance in the paper.

l155 - Marginal distributions are not used in calculating joint entropies. Please move them where necessary.

Thanks for catching this, it is fixed.

S2.2 This section contains quite a bit of information that should not be part of the Methodology.

S2.1(S2.2 in the revision#1 draft) was intended to help our readers interpret and understand the results presented in the paper. Since our methodology to reach our conclusions is not only calculating the objective functions, but also interpreting them, we consider laying out the key information measures as part of our methodology. Moving it to other sections would disrupt the flow of information and message of the paper.

S2.2 - I do not think the word "measure" as in the title works in this context (as a noun). Consider using other term such as metrics.

We kept measures, as it is more in line with how it is referred to in information textbooks. To make it easier to read, we renamed the section "Understanding and visualizing the measures of information" to avoid ambiguity between uses of the word "measures" as a noun and as a verb.

l184 - Additive properties were never discussed. What are these?

This refers e.g. to equation 6, which tells us that in the total information measured by joint entropy, we account for redundancy by subtracting the total correlation from the sum of entropies. We now explicitly refer to eq. 6 and use the more accurate term *additive relations*

l185 - Please consider clarifying this line as for our previous discussion.

We now clarified that with maximally informative, we mean reducing total uncertainty about the set of outcomes, rather than focusing on a specific variable or decision problem:

*….lead to a maximally informative sensor network, which minimizes total remaining uncertainty about the outcomes at all potential locations.*

l213 - "However, there is no consensus on how to minimize redundant information". I think that there is no consensus on its definition rather than on its ways to minimise it. Most authors will agree that the only way to find the "optimal" result is to resort to full-enumeration, while others have used more efficient methods to get there, so the problem can actually be (realistically) solved. In particular, when revising Alfonso et al. (2010), we found out that WMP is a criteria for defining redundancy, while the way to solve the problem was resorting to NSGA-II. These differences have to be clarified, as they seem mixed-up in the document.

Thanks. We modified the text to clarify this. We meant there is no consensus on how to minimize redundant information in terms of minimizing total correlation or pairwise transinformation.

Added in L163:

However, there is no consensus on *whether to use total correlation or transinformation metrics* to minimize redundant information

Eq 7-10 - are framed as greedy-add, as they read to optimise the given metric given a set of previous iterations, and a new candidate. Therefore, its formulation is not suitable for its use in the greedy-drop. Would not be better just to have a definition of the objective function, and present separately the way to solve it?

We agree that these equations are formulated from a greedy add perspective, because some of the methods we compare with (WMP and minT) are inherently formulated that way. We agree that MIMR and maxJE could be formulated differently without referring to Fc and the greedy add, but that would make the comparison with WMP and minT more complicated and would also make the way we illustrate the relations in figure 1 harder to understand. We now explicitly state the connection of this formulation to greedy and, and that MIMR and maxJE could be easily modified for exhaustive optimization:

Added in L244:

*Fc is the station considered for addition to the current set in a greedy-add approach. This formulation was chosen to allow a uniform presentation between methods.The objectives for methods maxJE and MIMR can easily be modified to consider Fc as part of S, so that the objective function evaluates the entire network rather than one candidate station for addition. This allows greedy-add, greedy-drop and exhaustive search methods.*

l277 - "This approach can for example be useful in Alpine terrain, where relocating a sensor requires significant effort (Simoni et al., 2011)" I think this argument is relevant when physically changing the placement of the sensors, but I do not see it as an argument when testing potential locations while in the "design" phase.

Having done fieldwork for that project, SW realized that often deployments are not planned fully in advance, but adapted based on data collected. In that sense, the design phase can be somewhat fluid. Because of the work involved and size of the team, hiking time, etc. a fairly limited number of stations per field visit could be placed. The decision where to place the next station, or whether to remove a sensor

from one location and place it somewhere else could also be made with help of similar design algorithms as discussed here. The objective functions could be the same, but the constraints could be different.

l288 - optimal combination of ... Better to be specific here.

Thank you, we changed to: *optimal combination of selected sensors for each network size*

S2 - I think at the end of this section there is not a clear presentation of the experimental setup to test the hypothesis presented at the introduction of the paper. I think it would be a good idea to re-draft this section to clearly point out what is the methodology of the paper to support your conclusions, rather than mixing it with the literature review. A diagram always helps. Also, in this point it would be best to be (briefly) specific in the formulations, and not leave those in the references or annex (such as in S2.4), as this is the core of your document.

We overhauled the structure of the Methodology section. We moved the S2.1 (Choice of scope) to the introduction section. In the new Methodology structure, we first introduce basic information formulas ( to provide context). Then, We layout the core argument of our paper and arguments around other alternative objective functions explored in this paper. We then provide the two diagrams to help our reader interpret information metrics and figures presented in the result section.

l368-379 - I think belongs to the literature review, as it does not show or discuss any of the results. If it does, I didnt find it, so please help reader pointing that out.

L375 to L379 contains the core reasoning of our argument in this paper.  The L368 to L374 provides context for this paragraph to highlight the difference between our argument and the opposing argument. Since in this paper, part of our method is reasoning about the objective functions, and our result is a contrast between what we argue for and an apparent consensus in the literature, our results and discussion contain atypically many references. We think these are needed at this point in the paper.

l381-395 - Idem

Thanks. We added more citations to this part of the text to support our argument.

l406 - "efficiency = bits of unique info / bits collected" -> efficiency (i.e. bits of unique information / bits collected)

We fixed as recommended.

l406 - bits collected -> bits

We fixed as recommended.

l423 "when increasing network size by one station" I think this should be removed. If we consider increasing the network by one element at the time then is greedy add and not exhaustive search. I think this will be clarified when the methodology shows that you will be testing the results for the optimisation of the network using 3,4,5.... n sensors.

We included this to highlight the subtle difference between increasing the network size by one station, and adding one station. We agree that this may add confusion and can be left to later to clarify, so we removed as recommended.

l428-430 - Don't downplay yourself or your setup :). It is well known that combinatorial problems are not simple to solve when they grow. However, try to quantitatively report what the results or setup is (type of PC, OS, etc).

Thanks :) As per previous reviewer comments, the most important point here is the exponential growth and the resulting ratio between 4 minutes and 5 years, which apply regardless of the other slowing factors. We think the actual setup is less relevant given the other mentioned inefficiencies.

l443-445 - I think this may go in the further work as is not something that is tested in this study.

Agree, we  removed it here as similar points were already made in the future work section.

l446 - Is well known that only exhaustive search can guarantee optimality. I think you can show this from well known (textbook) optimisation material.

We agree that exhaustive approaches are the only ones guarantee optimality in the general case of optimization. However, for a given specific problem, more efficient algorithms that guarantee optimality could in principle exist (e.g. dynamic programming for sequential decision processes). For this specific problem, it is important how information measures in the objective function interact. In this paper we empirically show from the result of exhaustive optimization that for joint entropy, no greedy optimal algorithm can exist. Maybe this could be shown from textbook material, but it would also need to include information theory textbooks then, since it could be specific to the objective function.

l446 - If there is a second dataset used in obtaining the results, this should be mentioned in the case study. Also, it has to be shown in the methodology what kind of experiments were carried out with it.
We did not use a second dataset in this paper. We randomly drew data from our original dataset to generate an artificial dataset. We performed this to show reaching the same results by exhaustive and greedy algorithms was not a general truth, and can be changed if we permute the data. The reason we introduce the artificial data is hard to justify beforehand, as the experiment was conceived in reaction to the initial result.

l458 - by definition, every search algorithm is more efficient than exhaustive search.
We agree with you. We modified the text and removed the comparison to the exhaustive algorithm.

T5 - This table reads way better now. Update the caption as is not monitor order, but selected stations. In addition, you can refer that you are using dataset #2 for these results if presented earlier in the case study and methodology sections. Finally, Make sure to use the same term (station) all over the paper.
We are glad you liked the updated version. Thank you for catching this. The exhaustive search indeed does not give a ranking, but a series of sets. We used the same dataset, but we randomly permuted the time series to generate an artificial dataset. We modified the Table's caption to reflect this.

T6 - Same comments as with T5. Please rename "Multivariate dimensions" to number of stations.
Was fixed as recommended.

l480 - Going back to our original discussion, I would like you to clarify that there is no justification on minimizing redundancy as long as there is no specific objective that supports that requirement. This is to highlight the point that (ideally) objective functions are created as a requirement for the optimisation, but given the fact that the problem is not that well defined in many applications, opting for joint entropy makes sense.
Our argumentation is still that there is no clear justification given for minimizing redundancy in previous literature, except for intuitive reasons that are already taken into account in joint entropy. We cannot think of a realistic scenario where minimization of redundancy would be warranted. Therefore in principle our advice is not to use it, also given the potential benefits of maximizing redundancy. If future research presents a clear case for why it should be included, we would be interested in reading that, but for now we do not want to speculate on that.

l489 - The dimensionality that feasible depends not only on the amount of stations, but also on the quantisation method, the potential superset of potential locations, and the hardware availability. I can imagine someone with an HPC cluster and a well-tuned implementation can process these results. I think it'll be better to leave it open on the side that complexity exponentially grows with the size of the problem and that should be considered.
We agree that the bound was not strong enough. Since the effect of exponential growth is sometimes difficult to commit to intuition, we think it is still helpful to illustrate with an example rather than leave open.

Since it is always enjoyable to read references to computational power back decades later, we included the following up to date reference point:

Added in L414:

[revised manuscript text omitted]